# Impacts of Climate and Land-Use Changes on Hydrological Processes of the Source Region of Yellow River, China

Mudassar Iqbal [1], Jun Wen [2],*, Muhammad Masood [1], Muhammad Umer Masood [1] and Muhammad Adnan [3]

1. Centre of Excellence in Water Resources Engineering, University of Engineering and Technology, Lahore 54890, Punjab, Pakistan
2. College of Atmospheric Sciences, Plateau Atmosphere and Environment Key Laboratory of Sichuan Province, Chengdu University of Information Technology, Chengdu 610225, China
3. Institute of International Rivers and Eco-Security, Yunnan University, Kunming 650500, China
* Correspondence: jwen@cuit.adu.cn

**Abstract:** Climate variability and land-use change are key factors altering the hydrology of a river basin, which are strongly linked to the availability of water resources and the sustainability of the local ecosystem. This study investigated the combined and individual impacts of land-use changes (LUCs) and climate change (CC) on the hydrological processes in subbasins of the Source Region of the Yellow River (SRYR) through statistical methods and hydrological modeling based on two land-use maps for the period 1990 and 2010, and two climate periods, i.e., 1976–1995 and 1996–2014. The results revealed that the climate is anticipated to be warmer and wetter. Land-use changes were dominated by decreases in sparse grassland. However, the transformation of land-use changes varied spatially within sub-basins. The combined impacts of climate and land-use changes are more noticeable in the Maqu subbasin, where the decrease in runoff reached 18% (32.2 mm) and increase in evapotranspiration (ET) reached 10.4% (34.1 mm), followed by the Tangnaihai and Jimai subbasins. The changes in runoff and ET caused by LUC seemed to be adequate by comparison, and presented a 15.1–19.8% decline in runoff and 21.3–28% increase in ET relative to the totals. Overall, climate change has more influence on hydrological processes in all subbasins of the SRYR than LUC. It is, therefore, considered that the response to changes in hydrological processes in a subbasin can be attributed to changes in individual climate parameters and land-use classes.

**Keywords:** Source Region Yellow River; climate change; land-use change; grassland; runoff; evapotranspiration; soil and water assessment tool

## 1. Introduction

It is globally agreed that the main drivers of climate change (CC) and land-use change (LUC) affect basin hydrology, which has led to the recognition of the need for effective water management and conservation of fluvial ecosystems at basin scale [1–3]. CC, with raised temperature and altered precipitation intensity and patterns, causes variations in hydrological regimes such as low flows, peak flows and time of flow routing [3–8]. Land-use changes also affect hydrological processes such as interception, infiltration and evapotranspiration [9–12], causing variations in surface and subsurface flows. This is mainly true in arid and semi-arid areas, where the ecosystems are fragile and sensitive to climate change, and the scarcity of water significantly constrains economic and ecosystem developments. The changes in hydrological processes in the source areas of rivers in the aforementioned regions ultimately affect the water yield in downstream reaches. Thus, the assessment of impacts by CC and LUC have garnered considerable attention from the scientific community, as well as decision-makers, for the optimal management of water resources from the perspective of increasing water scarcity [13,14].

Studies have been conducted to examine climate change's effects on hydrological components, and its parameters role has been identified [15–20]. Scientists have also

paid attention to the combined impacts of climate change and LUC on the hydrological components of the river basin [1,14,16,21]. The majority of studies used hydrological modelling or statistical modelling to assess changes in hydrological processes at the basin scale [17,21,22]. However, some of the research deals with the individual impact of climate change and LUC and presented different results for different basins. For example, Chung et al. [17] reported that variations in streamflow are remarkable because of climate change in the Anyangcheon basin, Korea. In semi-arid region, the impacts of LUC are more remarkable than those of climate change [18,19]. Daofeng et al. [20] presented the impacts of CC and LUC on streamflows in the Yellow River's upper reaches, where climate change had a greater influence than LUC. However, studies conducted on the Loess Plateau as well as Yangtze River delta region in China also concluded that the changes in streamflow are mainly caused by LUC [21–23]. Since the dominant factor, LUC or climate change, varies spatially due to topographical conditions and geographical varieties, further research is needed to conserve the eco-environment of a particular basin.

The Yellow River, Mother River of China is the second longest river in China. As a source of the Yellow River, eco-systems of source region of the Yellow River (SRYR) have a significant water resource recharging function that determines the water availability in the river. Since the region is characterized by a complex orography and fragile ecosystem [17,24,25], the water yield is susceptible to average and increased variations in climate change and LUC [17–20,22,23]. The major ecosystem covered by alpine meadows has been degraded in the region [26–28]. Moreover, through the implementation of Ecological Protection and Restoration Program (EPRP) in 2005, the Normalized Difference Vegetation Index (NDVI), together with temperature and precipitation, has increased in three headwater regions of China. The increment in grassland was more pronounced in the southeast of the region [29]. There were degradations in the grassland area, accounting for 36.12%, which resulted in a substantial impact on water-retaining capacity [30]. Specifically, due to grassland degradation, water retention decreased in low-lying areas and the freeze–thaw process declined in high-altitude areas [25]. Moreover, the trend of observed flows in the SRYR or other regions decreased, although the actual source of this decrease is still unclear due the differing views in studies. Zhang et al. [31] and Daofeng et al. [20] suggested that climate change was the major reason of these changes in streamflow in the source of the three rivers in China and the SRYR, respectively, while the streamflow changes were primarily attributed to LUC in the Loess Plateau of China by Zou et al. [22] and Yan et al. [23]. It was proved that the LUC and CC are the main drivers varying the water supplies in any region, as well as their spatial contributions. Therefore, there is a need to inspect the contribution of land-use and climate changes to the hydrological processes (runoff and evaporanspiration) in the SRYR, where the climatic [2] and land-use patterns vary at the subbasin scale [9].

Numerous studies have been conducted to examine the impacts of climate change and land-use changes on hydrological response in the SRYR, and they primarily emphasize two concerns: (i) past or future climate change and its impacts [5–7,9,16,27]; (ii) past or future hydrological impacts of land-use changes and climate variability [11,20,28,29]. The majority of these studies consider the hydrological variation looking only at climate change, and few studies have analysed the combined impacts. In addition, the runoff variation was evaluated statically [5–7,16], instead of using an integrated approach with a hydrological model. Since the applied statistical methods were not physically explicit, the quantification of impact caused by climate or land-use will not be easy. Moreover, the climate or/and land-use data either had a coarse resolution [11,32] or were not the most recent. Some of the studies analysed the combined hydrological impacts, and climate change scenarios were established using global climate model (GCM) projection data. For example, GCM multi-model ensemble projections were used under three emissions scenarios, A1B, A2 and B1, for the near future, by Zhang et al. [27]. Wang et al. [28] determined the impacts on streamflows in the SRYR looking at future climate-change scenarios using three representative concentration pathways (RCPs) (RCP2.6, RCP4.5, and

RCP8.5) with a land-use scenario for 2025. However, the exploration of hydrological variation in hypothetical scenarios with the combination of temperature and precipitation and five land-use scenarios [20], and development of different sets of climate data with respective land-use for the period of 1981–2005, as conducted by Pan et al. [11], seems more reliable than GCM climate scenarios when applied to a highly spatially heterogeneous and complex terrain, such as the SRYR, owing to the inherent coarse resolution of GCMs [33]. Additionally, the quantitative assessment of hydrological processes in response to climate and land-use changes were rarely explored by applying the integrated modelling approach.

In this study, we provide an update on the impacts of land-use and climate change on the hydrological processes by apply integrated modelling combined with the Soil and Water Assessment Tool (SWAT) and statistical framework proposed by Yang et al. [34]. The objectives are (a) to investigate the variation in climatic and hydrological parameters using the observed data from 1974 to 2014; (b) to characterize the changes in land-use and their attribution to changes in the hydrological processes. Overall, the results will provide beneficial knowledge, allowing for water-resource management to deal with the impacts that climate and land-use changes have on runoff and evapotranspiration at Tangnaihai, Maqu, and Jimai subbasins within the SRYR. To the best of our knowledge, this will be first comprehensive study at the subbasin scale.

*Study Area*

Figure 1 shows the geographical location of the SRYR, which is situated in the Qinghai-Tibet (QT) Plateau in China between 32°12′–35°48′ N and 95°50′–103°28′ E. The region has an area (121,972 km$^2$) that is covered by the upstream catchment area of the Tangnaihai hydrological station. The SRYR supplies of 35% of the total annual flow to downstream reaches; hence, it is called the water tower of the Yellow River Basin. The northern and southern parts of the region are enclosed by mountains, while the western side is characterized by highlands. There are three major subbasins in the SRYR, named Tangnaihai, Maqu and Jimai. The elevation in the SRYR ranges from 2666 m to 6253 m and falls from the west to the east. The more highly elevated areas of the region are mostly located in the Jimai subbasin; however, the highest change in elevation exists in the Tangnaihai subbasin due to the Anyamaqin Mountain peaks and low elevation area of the basin, such as the outlet of the region.

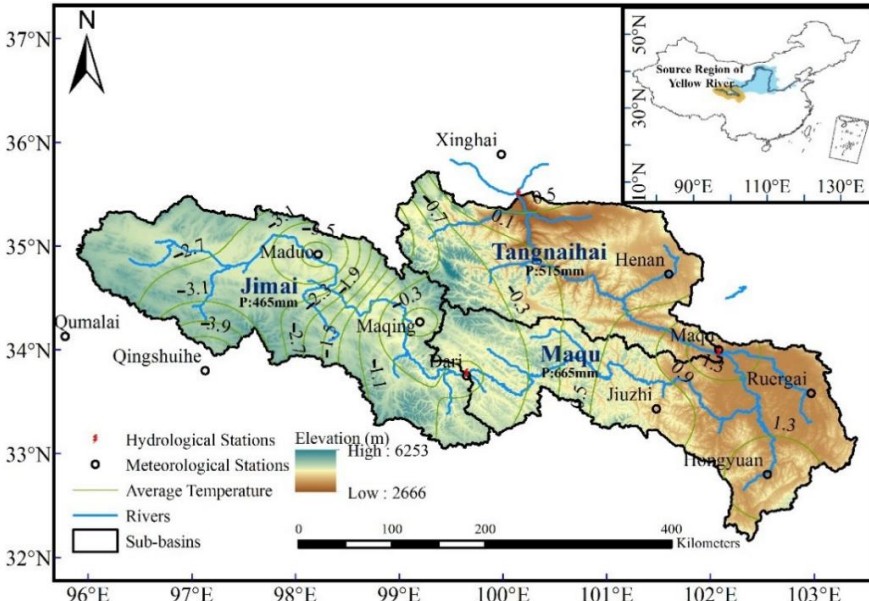

**Figure 1.** Location map of the SRYR. Maqu, Tangnaihai, and Jimai are represented by their respective black boundaries, and P stands for their combined average annual precipitation. The values of average temperature contours can also be shown in the map.

The region's land cover can be described as lakes/wetland, snow, glaciers, vegetation, and rivers. The vegetation is of a cold-tolerant perennial type, which covers the grassland (steppe and shrub meadows, alpine swamp) and forest (coniferous, broadleaf, needle leaf and mixed forest). Forest covers 11.4 percent of the area, while grassland, the primary ecosystem of the region, covers over 80 percent of the total land area [35]. However, grassland has been shown to degrade due to overgrazing and climate change [35,36]. The Tangnaihai subbasin has a permanent snow cover and 58 glaciers, representing 95.8% of the total glacier area (134 km$^2$) [9]. There are around 5300 lakes (2000 km$^2$) in the region, together with two major lakes, Ngoring (610 km$^2$) and Gyaring (550 km$^2$), which exist in the Jimai subbasin [17,34].

Climatically, the SRYR is characterized by a dry winter and wet summer; however, the climate conditions vary spatially. The annual average temp in the SRYR varies between 2.0 °C and −4.0 °C from the southeast to northwest of the region [7]. The southwest monsoon from the Indian Ocean influences the southern plateau, while the eastern plateau is influenced by the southeast monsoon from the Pacific Ocean [37]. The mean annual precipitation is 530 mm yr$^{-1}$, reaching its maximum in Maqu subbasin (665 mm yr$^{-1}$), while the minimum precipitation occurs in the Jimai subbasin (465 mm yr$^{-1}$). The mean annual duration of sunshine is 3028 h, the potential evaporation ranges between 800 and 1200 mm yr$^{-1}$, and the annual average temperature of the region increases from −4.0 °C to 2.0 °C from the orthwest to the southeast [7].

## 2. Materials and Methods

### 2.1. Dataset

#### 2.1.1. Hydro-Meteorological Data

The list of topographic, hydrological and climatic data used in this study is given in Table 1. The China Meteorological Administration provided the meteorological data for average, minimum and maximum temperatures, precipitation, sunshine hours, relative humidity and wind speed of eleven stations for the period 1974–2014. Streamflow records of the hydrological stations (Tangnaihai, Maqu, and Jimai) were collected from the Yellow River Conservancy Commission for the same period. To ensure the quality and homogeneity of time series data, standard statistical approaches were employed and their results are discussed in previous studies [2,7,38,39].

**Table 1.** The catalogue of datasets utilized in this research.

| Category | Data Set | Source |
|---|---|---|
| Meteorology | Relative humidity, temperature, precipitation, sunshine hours, wind speed | China Meteorological Administration, Yellow River Conservancy Commission |
| Topography | Digital elevation model (DEM) | Shuttle Radar Topographic Mission (version 4) (https://srtm.csi.cgiar.org/ (accessed on 15 March 2018)) |
| Hydrology | Discharge | Yellow River Conservancy Commission |
| Soil | Digital soil maps of China | Harmonized World Soil Database (version 1.2) (http://webarchive.iiasa.ac.at/Research/LUC/External-World-soil-database/ (accessed on 20 March 2018)) |
| Land-use | Current land-use data of China | Landsat TM/ETM remote sensing images processed by the Data Center for Resources and Environmental Sciences, Chinese Academy of Sciences |

#### 2.1.2. Topographic Data

The Digital Elevation Model (DEM) of 90 m spatial resolution was downloaded from the Shuttle Radar Topographic Mission (SRTM) website in Geo TIFF format (Table 1). The subbasins boundaries for Tangnaihai, Maqu and Jimai were delineated using ArcHydro tools in an Arc GIS environment (Figure 1).

### 2.1.3. Land-Use Data

Land-use maps (1:100,000) of the 1990s and 2010s were requested from the Resources and Environmental Sciences Data Centre (RESDC), Chinese Academy of Sciences. Previously, land-use data were interpreted based on the Landsat Thematic Mapper (TM), to identify ecological land uses [30,40]. The land-use change (LUC) data comprised six major types, which were further divided into 25 subtypes. The six major LUC types included croplands, woodlands, grasslands, water, construction land, and unused land. (i) The croplands comprised dry land and paddy; (ii) the forests comprised sparse woodland, shrubland, and other woods; (iii) the grasslands contained low-coverage, moderate-coverage, and high-coverage grassland; (iv) the construction land included urban and rural areas and settlements; (v) the water included lakes, canals, reservoirs, ponds, snow and ice, shoals, and beaches; (vi) the unused land included desert, sandy land, salty land, bare ground, and marsh. The LUC data were further reclassified into eight different types, and it was found that the grassland area was increased by 3.42%, whereas the sparse grassland area was reduced by 4.63% in 2010 compared to 1990 for the SRYR, as shown in Figure 2.

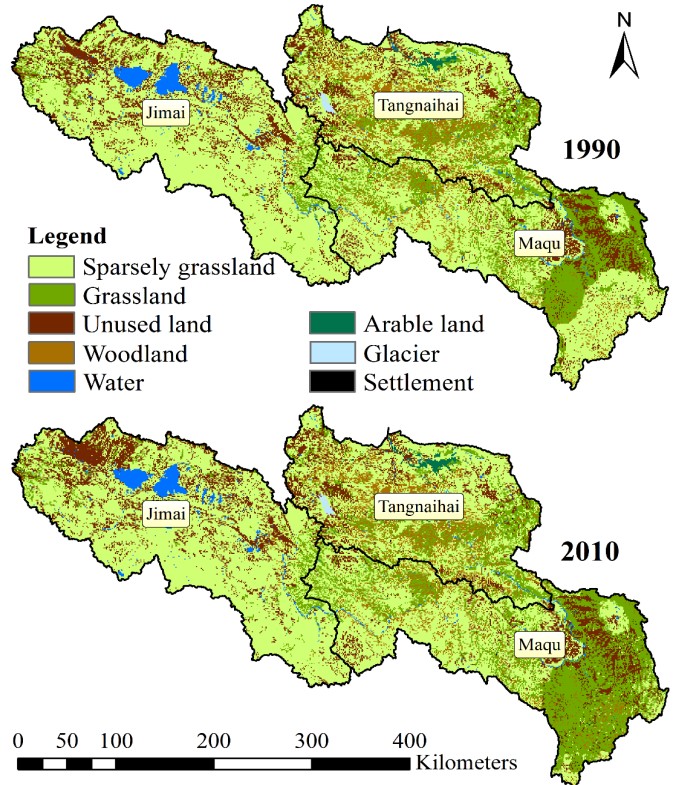

**Figure 2.** Land-use classification for 1990s and 2010s in the SRYR.

### 2.1.4. Soil Data

Food and Agriculture Organization (FAO) and International Institute for Applied Systems Analysis provide the Harmonized World Soil Database V 1.2 (HWSD), which is 30 arc-second database [41]. The soil data are available at the scale of 1:1,000,000. A soil map of China was used to extract the study area. Each soil unit of HWSD consists of organic carbon content, depth, electrical conductivity, bulk density and soil texture, which can be directly utilized in the soil database of the Soil Water Assessment Tool.

The values of a few important factors, including saturation, field capacity, saturated hydraulic conductivity, wilting point and available water, were determined using the soil-plant–atmosphere–water (SPAW) model. The calculated parameters were based on the texture of soil, as given in the soil database. Gelic Leptosols was found to be dominant in the SRYR, followed by Mollic Leptosols. Gelic properties refer to soils having permafrost within

200 cm of the soil surface. Mollic lepotols refers to the soils that contain or immediately over-lie a calcium carbonate content of greater than or equal to 40%.

Due to its rough terrain and vast area, the SRYR was classified into five slope classes: 0–5%, 5–25%, 25–48%, 48–75%, and >75%. Of these, 43% of the SRYR was determined to have a slope of within 5–25%.

### 2.2. Methods

#### 2.2.1. Trend Analysis

Sen's slope method and the Mann–Kendall [42,43] trend analysis were performed on the hydro-meteorological data of the SRYR to estimate the increasing and decreasing trends and their magnitudes. The Mann–Kendall test is a rank-based, non-parametric test. It is not dependent on the data being properly distributed and is less sensitive to inhomogeneous time series [44]. The details of Sen's slope method and Mann–Kendall are provided in Safeeq et al. [45].

#### 2.2.2. Hydrological Modeling

The modelling was performed using the Soil and Water Assessment Tool (SWAT) developed by Arnold et al. [46]. The SWAT is a physical-based, semi-distributed hydrological model with the capacity to predict the influence of land-management practices on water, water-quality parameters and sediment yield in intricate basins with varying land-use and soil conditions at annual, monthly and daily scales. For example, in the SRYR, the main hydrological component (runoff) was mainly contributed by precipitation (rainfall and snow). The contribution from glacial melt is negligible because of its minute coverage of the total area (0.11%) [9]. The SWAT model is inherently able to deal precipitation to runoff processes, meaning that it is applicable in the SRYR to evaluate the hydrological components. The SWAT model proved to be an effective tool for examining hydrological processes in response to varying climate and land-use changes [4,15,34]. The model divides a basin into a number of subbasins based on hydrological response units (HRUs). HRU is a region that exhibits a unique configuration of elevation, soil type and land-use. The model estimates the water balance for each HRU and accumulates the HRUs at the outlet of the basin. In this study, hydrological components are simulated at each subbasin by applying the water balance equation, as provided below [46]:

$$SW_t = SW_0 + \sum_{i=1}^{t} (R_{day} - Q_{surf} - E_a - W_{seep} - Q_{gw}) \tag{1}$$

where $Q_{gw}$ (mm) represents return flow on the $i^{th}$ day, $W_{seep}$ (mm) describes the volume of water that entered the vadose zone on day one from the soil profile, $E_a$ (mm) represents evapotranspiration on the $i^{th}$ day, $Q_{surf}$ (mm) is the runoff on $i^{th}$ day, $R_{day}$ (mm) denotes the precipitation on $i^{th}$ day, $SW_o$ (mm) specifies the initial soil moisture, and $SW_t$ (mm) represents the final soil moisture.

The SWAT model estimates surface runoff ($Q_{surf}$) using the SCS curve number method [47] and Green-Ampt Infiltration method [48]. In this study, the SCS curve number method was chosen to compute surface runoff by the following equation:

$$Q_{surf} = \frac{\left(R_{day} - 0.2S\right)^2}{\left(R_{day} + 0.8S\right)} \tag{2}$$

where $Q_{surf}$ (mm) is excess rainfall or surface runoff, $R_{day}$ (mm) is the rainfall in a day and $S$ (mm per day) is the maximum retention parameter. This parameter varies spatially

and temporally due to variations in land-use management, slope and soils, and soil water content, respectively. The retention parameter equation is given by:

$$S = 25.4\left(\frac{1000}{CN} - 10\right) \tag{3}$$

where *S* is defined earlier and CN denotes the curve number.

Model Setup

The SWAT model version, i.e., SWAT2012, was applied to the Tangnaihai, Jimai and Maqu subbasins of the SRYR. The model was driven by topography, soil land-use and meteorological data [46,49]. The model delineated 34 subbasins for the whole of the SRYR based on the 150,000 hectare areas, which include hydrological stations, namely, Maqu, Jimai, and Tangnaihai. To consider the orographic effects and better depict the distribution of meteorological parameters over subbasins, each subbasin was split into ten bands based on elevation. The study by Zhang et al. [27] determined that the precipitation and temperature lapse rates should be set at +0.5 mm/km and −6.5 °C/km, respectively. The homogeneous type of slope, land-use, and soil were used to further classify the subbasins into 480 HRUs. The hydrological processes were preceded using the SCS-CN method, while the Muskingum scheme was chosen for channel flow routing and the Penman–Monteith equation was selected to estimate the evapotranspiration.

Calibration and Validation of Model

The Sequential Uncertainty Fitting Method (SUFI-2) algorithm was adopted for calibration and sensitivity analysis using SWAT calibration and uncertainty programme (SWAT-CUP) [48]. The parameters of the SWAT-CUP model were adjusted based on actual streamflow measurements made at the outlets of the Tangnaihai, Maqu and Jimai subbasins. Parameters were initially sorted from the studies conducted over the whole Yellow River Basin and the SRYR [5,27,50]. To complete the sensitivity analysis, each parameter within the range was also altered, while the other parameters were kept at realistic levels. The list of calibrated parameters is presented in Table 2. The sensitive parameters for the SRYR were the curve number, bulk density of the soil layer, saturated hydraulic conductivity, the groundwater delay time, deep aquifer percolation fraction, snowpack temperature lag factor, melt factor for snow on June, snowmelt base temperature, melt factor for snow on December 21, snowfall temperature, average slope steepness, and soil evaporation compensation factor. The model was trained for three years (i.e., 1973–1975) as a warm-up period, with the climate forcing data to alleviate the effects of imprecise initial conditions. The SWAT model was jointly calibrated from 1976 to 1995 using land-use for the year 1990, and validated for the period from 1996 to 2014 with the land-use data for 2010 against the observed streamflows at the Jimai, Maqu, and Tangnaihai hydrological stations.

Performance Evaluation of Hydrological Model

The coefficient of determination ($R^2$) and the Nash–Sutcliffe efficiency (NSE), the two statistical indicators suggested by Moriasi et al. [50] were used to assess the model performance. The performance of the model during the calibration and validation stages was also measured using the percentage bias (PBIAS) [51]. Their respective equations are as follows:

$$R^2 = \left[\frac{\sum_{i=1}^{n}(Q_o - \overline{Q}_o)(Q_s - \overline{Q}_s)}{\sqrt{\sum_{i=1}^{n}(Q_o - \overline{Q}_o)^2 \sum_{i=1}^{n}(Q_s - \overline{Q}_s)^2}}\right]^2 \tag{4}$$

$$\text{NSE} = 1 - \frac{\sum_{i=1}^{n}(Q_0 - Q_s)^2}{\sum_{i=1}^{n}(Q_0 - \overline{Q}_o)^2} \tag{5}$$

$$PBIAS = \frac{\sum_{i=1}^{N}(Q_o - Q_s)^2}{\sum_{i=1}^{N}(Q_o)} \times 100\% \tag{6}$$

where $n$ denotes the number of datapoints; $Q_s$ and $Q_o$ are the simulated and observed runoff at $i^{\text{th}}$ time step; $\overline{Q}_s$ and $\overline{Q}_o$ denote the simulated and observed runoff, respectively.

**Table 2.** Final optimized values of calibrated parameters for the SRYR.

| Parameter | Description | Minimum Value | Maximum Value | Fitted Value |
|---|---|---|---|---|
| v__GW_DELAY.gw | Groundwater delay (days) | 64.85 | 154.97 | 105.72 |
| v__SMFMN.bsn | Minimum melt rate for snow during the year (winter) $H_2O/°C$-day) | 4.91 | 14.97 | 7.58 |
| r__SOL_K.sol | Saturated hydraulic conductivity (mm/h) | −1.16 | 0.15 | −0.58 |
| r__SOL_BD.sol | Moist bulk density (g/cm$^{-3}$) | −0.02 | 0.94 | 0.73 |
| v__RCHRG_DP.gw | Percolation from deep aquifer | 0.20 | 0.73 | 0.48 |
| v__TIMP.bsn | Snowpack temperature lag factor | −0.02 | 0.66 | 0.05 |
| v__SMTMP.bsn | Snow melt base temperature (°C) | −2.61 | 2.47 | 0.69 |
| v__SFTMP.bsn | Snowfall temperature (°C) | −6.99 | 1.01 | −0.53 |
| r__CN2.mgt | Initial SCS runoff curve number for moisture condition II | −0.70 | −0.1 | −0.56 |
| v__HRU_SLP.hru | Average slope steepness (m/m) | 0.46 | 1.39 | 0.90 |
| v__ESCO.hru | Soil evaporation compensation factor | 0.49 | 1.46 | 0.95 |
| v__SMFMX.bsn | Maximum melt rate for snow during the year (mm $H_2O/°C$-day) | 8.01 | 24.03 | 10.13 |

### 2.2.3. Statistical Framework for Individual Impact of Land-Use Changes and Climate Change

LUC and climate change each have a unique effect on hydrological processes and have been widely studied using the one-factor-at-a-time (OFAT) approach. However, the impact of each cannot be quantified with a conventional OFAT approach because the total contribution of each factor does not make 100%. For example, Li et al. [51] identified the impacts of CC (95.8%) and LUC (9.6%) on streamflow in the Loess Plateau of China during the 1981–2000, which appeared to be decreasing. The primary drawback of the conventional OFAT approach is that, while evaluating the impact of one factor on the hydrological process over a particular study period, the other factors' effects are not considered. Therefore, studying the individual impacts of two factors (CC and LUC) on hydrological processes requires the applicable status of each factor from the reference period to the entire period to be input, as considered in the modified OFAT proposed by Yang et al. [14]. Recent researcs on other watersheds found the modified OFAT proposed by Yang et al. [14] to be efficient [3,4,15], and the same method was adopted. In this study, climatic data were divided into two periods: the first (C1) was used as a baseline period and the second one (C2) was used as a comparison period. The land-use conditions are defined for both periods by two land-use maps, L1 and L2, respectively. The configurations of C1, C2, L1, and L2 are often used to compute the integrated modelling scenarios (SI, S2, S3, and S4). Figure 3 shows a schematic representation of this method and the changes in runoff and ET with climate change variables under various land-use maps. The hydrological fluctuations caused by the CC and LUC were calculated using Equations (7)–(9), and their combined impacts, respectively, are as follows:

$$\Delta Q_L = \frac{1}{2}\left[(Q_{C1}^{L2} - Q_{C1}^{L1}) + (Q_{C2}^{L2} - Q_{C2}^{L1})\right] \tag{7}$$

$$\Delta Q = Q_{C2}^{L2} - Q_{C1}^{L1} \tag{8}$$

$$\Delta Q_C = \frac{1}{2}\left[(Q_{C2}^{L1} - Q_{C1}^{L1}) + (Q_{C2}^{L2} - Q_{C1}^{L2})\right] \tag{9}$$

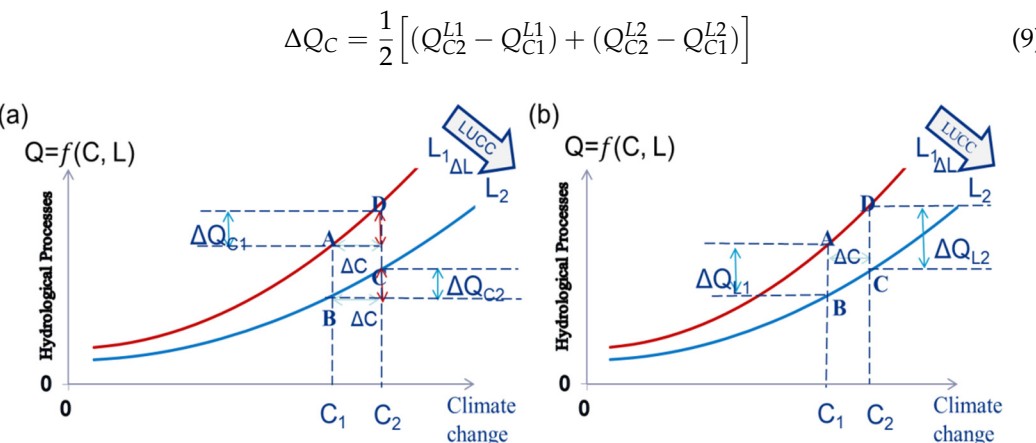

**Figure 3.** A conceptual diagram of the individual impacts under different scenarios on runoff/ET: by climate change (**a**) and land-use change (**b**).

This study employed land-use maps from the 1990s and 2010s and meteorological data from 1976–1995 and 1996–2014 to reveal the land-use patterns of each. Each of the four options derived from these two land-use maps and climatic eras was subjected to a calibrated SWAT model. Here, these are identified as four distinct situations. By comparing the results of four SWAT model scenarios, the effects of LUC and CC may be evaluated (Table 3).

**Table 3.** Scenarios to separately quantify the impacts of CC and LUC on runoff using calibrated SWAT model.

| Scenario | Period of Land-Use Map | Period of Climatic Data |
|----------|------------------------|-------------------------|
| S1 | 1990s | 1976–1995 |
| S2 | 2010s | 1976–1995 |
| S3 | 1990s | 1996–2014 |
| S4 | 2010s | 1996–2014 |

## 3. Results

### 3.1. Decadal Changes in Hydro-Meteorological Variables of the SRYR

Annual mean monthly temperature, precipitation and runoff, as hydro-metrological variables, were averaged on the basis of daily and monthly data series. The results of annual trends per decade and decadal means for the period 1974–2014 are shown in Figure 4. Over the course of entire period, temperature and precipitation exhibited significant and insignificant increasing trends at the rate of 0.4 °C per decade and 2.3 mm per decade, while runoff presented an insignificant decreasing trend at the rate of 36.3 m$^3$/s per decade. It is evident from decadal means that the regional temperature will continuously increase, with a maximum mean of 0.29 °C in the last decade (2004–2014). Runoff following the decadal average pattern of precipitation was exhibited to decrease from the first to third decades, over the period 1974–1994. In the last decade (2004–2014) the mean values of precipitation and runoff increased. However, the mean runoff of the last decade is less than that of the first, which may result in a significant temperature change in the last decade (0.29 °C), which was quite high compared to the first decadal mean temperature (−0.9 °C).

The Mann–Kendall test was performed to determine the significance of the trends, as shown in Figure 5, to ensure statistical validity. This analysis showed different trends in three variables, as well as statistically significant trends for certain months over the entire period (Table 4). The precipitation trends were insignificant in all months except May, where a significant upward tendency in precipitation, of 3.39 mm per decade at a

confidence level of 10%, was observed. The substantial positive spikes in precipitation from May to July may lead to increases in annual precipitation.

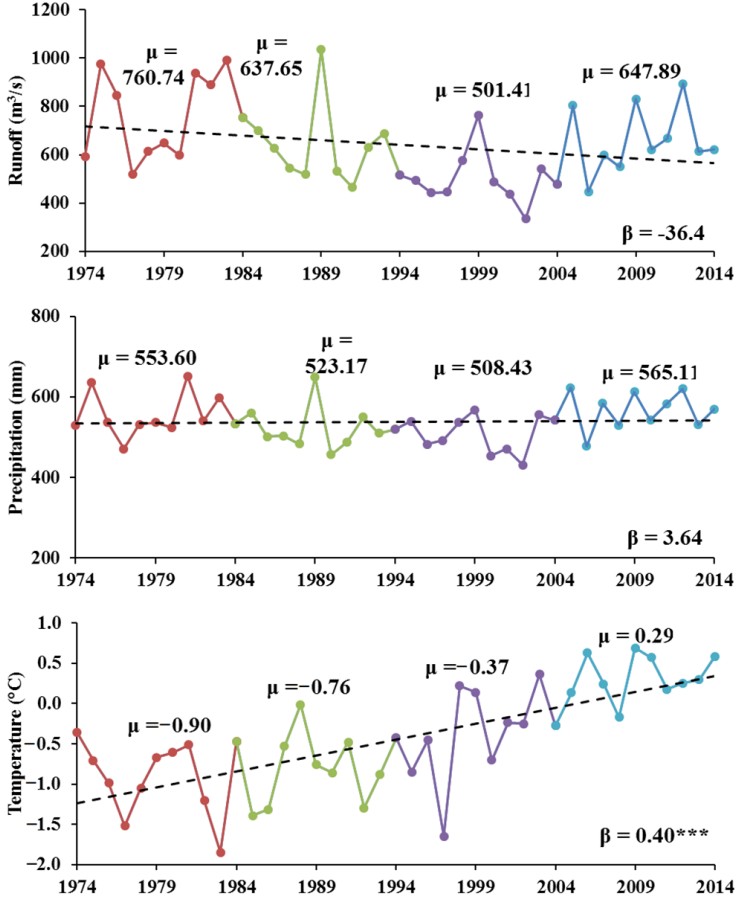

**Figure 4.** Sen's slopes (β, unit: per decade) of annual runoff, precipitation, and temperature for the whole period 1974–2014. μ represents the respective decadal mean of the variables and red, green, purple, and blue colors show the different decades. The significance of the trend at $p < 0.001$ is indicated triple asterisk sign.

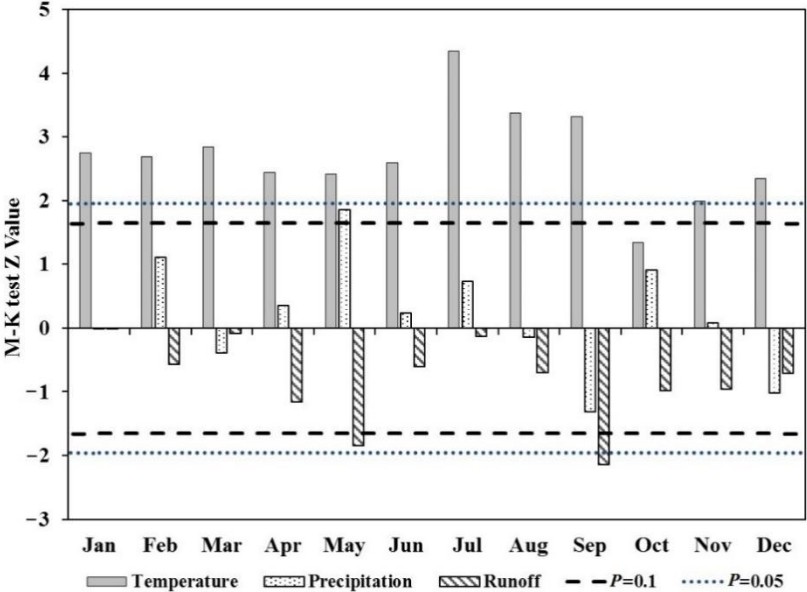

**Figure 5.** Monthly Mann–Kendall test results for runoff, precipitation, and temperature.

**Table 4.** Sen's slope (per decade) and the MK tests are used to analyze trends in hydrometeorological variables.

| Month | Temperature | | | Precipitation | | | Runoff | | |
|---|---|---|---|---|---|---|---|---|---|
| | Test Z | Sig. | β | Test Z | Sig. | β | Test Z | Sig. | β |
| January | 2.75 | ** | 0.59 | −0.01 | | −0.02 | −0.01 | | 0.00 |
| February | 2.68 | ** | 0.70 | 1.11 | | 0.51 | −0.57 | | −2.71 |
| March | 2.84 | ** | 0.38 | −0.39 | | −0.50 | −0.08 | | −0.44 |
| April | 2.44 | * | 0.31 | 0.35 | | 0.44 | −1.16 | | −13.76 |
| May | 2.41 | * | 0.28 | 1.85 | + | 3.39 | −1.84 | + | −41.63 |
| June | 2.59 | ** | 0.30 | 0.24 | | 1.09 | −0.61 | | −20.90 |
| July | 4.35 | *** | 0.55 | 0.73 | | 3.15 | −0.13 | | −10.31 |
| August | 3.38 | *** | 0.42 | −0.15 | | −0.45 | −0.70 | | −39.21 |
| September | 3.31 | *** | 0.52 | −1.31 | | −3.76 | −2.15 | * | −143.67 |
| October | 1.34 | | 0.18 | 0.91 | | 1.33 | −0.98 | | −44.92 |
| November | 1.99 | * | 0.24 | 0.08 | | 0.03 | −0.95 | | −16.70 |
| December | 2.35 | * | 0.42 | −1.02 | | −0.27 | −0.71 | | −5.77 |

Note: Plus, single, double and triple asterisk signs indicating the significance of the trends at $p < 0.10$, $p < 0.05$, $p < 0.01$ and $p < 0.001$, respectively.

The statistics on air temperature showed a warming trend over the whole year. Except for October, the increasing trends were significant at different confidence levels (Table 4). The highest rate, of 0.70 °C/decade, was observed in the month of February. The average rate between June and August (0.42 °C/decade) was 1.3 times slower than the average between December and February (0.57 °C/decade). The warming tendency was more pronounced during the winter months compared to the summer months.

The runoff trends showed a decreasing tendency during all the months, but appeared to be significant in May and September. Between April and November, variations occurred in the trends in monthly runoff, and this fluctuation persisted throughout the whole period (Table 4). These variations showed positive responses to the variation in precipitation, which were not subject to delays over the following months. Interestingly, the variations in precipitation and runoff over the monthly period occasionally reached statistical significance, reflecting trends that could be seen in both yearly and monthly scales runoff and precipitation data (Figures 4 and 5).

### 3.2. Changes in SRYR Land-Use

Figure 2 and Table 5 indicated changes in land-use composition in the SRYR that have occurred since the early 1990s. The dominating land-use class was sparse grassland, followed by grassland, which covered 56% and 15% of the entire expanse, respectively. An increase in grassland, settlements, unused land, woodland and water, and arable land was observed, and only sparse grassland showed a decrease. Overall, during the study period, the grassland in the SRYR area increased and the environment improved.

Table 6 showed the land transformation from the 1990s to 2010s, which comprised 11,614 km², 9.7% of the total area. In this study, sparse grassland, grassland, woodland, and unused land remained under special consideration, as they may have substantial effects on various hydrological processes. Analysis showed that grassland mainly increased due to the changes in sparse grassland (7790 km²), unused land (1621 km²), and woodland (1406 km²), all a result of the ecological restoration programs launched in the early 2000s. Unused land increased due to changes from grassland (1505 km²) and into grassland (1621 km²). Figure 6 shows major transformations in specific subbasins of the SRYR and Table 7 shows the predominating form of these changes. Changes in a certain pair of

land-use classes depend on net transformations from and into other land-use classes. More than 10 km$^2$ of total land-use underwent nine changeovers in two categories: one is sparse grassland to other types of land-use and the second is other types of land-use to grassland and water.

**Table 5.** Comparison of land-use changes in the SRYR during the 1990s–2010s.

| Land-Use Type | 1990 | | 2010 | | Change | |
|---|---|---|---|---|---|---|
| | Area (km$^2$) | Area (%) | Area (km$^2$) | Area (%) | Area (km$^2$) | Area (%) |
| Settlements | 33.00 | 0.03 | 44.00 | 0.04 | 11.00 | 0.01 |
| Glaciers | 157.00 | 0.13 | 157.00 | 0.13 | 0.00 | 0.00 |
| Arable land | 394.00 | 0.33 | 469.00 | 0.40 | 75.00 | 0.06 |
| Water | 2281.00 | 1.93 | 2345.00 | 1.98 | 64.00 | 0.05 |
| Woodland | 8543.00 | 7.21 | 8662.00 | 7.31 | 119.00 | 0.10 |
| Unused land | 16,573.00 | 13.99 | 17,747.00 | 14.98 | 1174.00 | 0.99 |
| Grassland | 18,248.00 | 15.40 | 22,311.00 | 18.83 | 4063.00 | 3.42 |
| Sparse grassland | 72,238.00 | 60.98 | 66,771.00 | 56.34 | −5467.00 | −4.63 |

**Table 6.** Land-use change matrices in the SRYR from 1990 to 2010.

| Transformation Matrix | | The 2010s | | | | | | | |
|---|---|---|---|---|---|---|---|---|---|
| | | Sparse Grassland | GrassLand | Unused Land | WoodLand | Water | Arable Land | Glacier | Settlements |
| The 1990s | Sparse Grassland | 53,101 | 7790 | 7159 | 3488 | 551 | 106 | 20 | 19 |
| | Grassland | 4002 | 11,329 | 1505 | 1252 | 105 | 42 | 0 | 11 |
| | Unused land | 5855 | 1621 | 8691 | 184 | 176 | 28 | 16 | 2 |
| | Woodland | 3175 | 1406 | 196 | 3710 | 53 | 3 | 0 | 0 |
| | Water | 514 | 126 | 158 | 23 | 1451 | 7 | 0 | 2 |
| | Arable land | 67 | 19 | 21 | 0 | 5 | 280 | 0 | 2 |
| | Glacier | 27 | 0 | 9 | 0 | 0 | 0 | 121 | 0 |
| | Settlement | 11 | 8 | 3 | 0 | 0 | 3 | 0 | 8 |

Note: The units for changing land-use are in km$^2$.

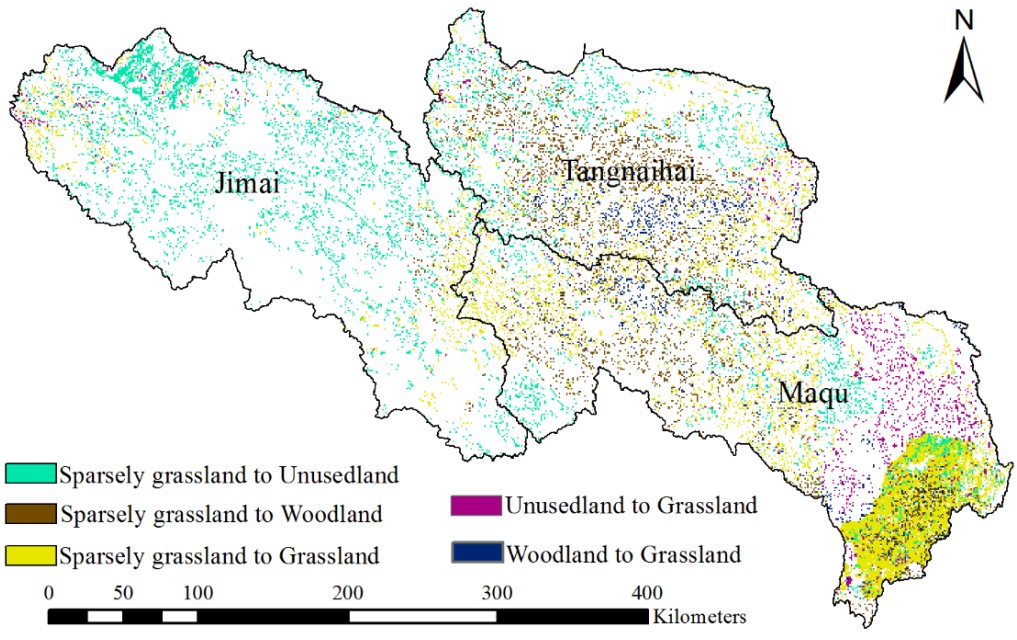

**Figure 6.** Spatial distribution of dominated land-use conversion in the SRYR.

**Table 7.** Changes in land-use in the SRYR catchment between 1990 and 2010.

| Predominating Transformation | The Net Change in Area (km²) | Change in Land-Use (%) |
|---|---|---|
| Unused land to water | 18.00 | 0.11 |
| Water to grassland | 21.00 | 0.92 |
| Grassland to arable land | 23.00 | 0.13 |
| Woodland to water | 30.00 | 0.35 |
| Unused land to grassland | 116.00 | 0.70 |
| Woodland to grassland | 154.00 | 1.80 |
| Sparse grassland to woodland | 313.00 | 0.43 |
| Sparse grassland to unused land | 1304.00 | 1.81 |
| Sparse grassland to grassland | 3788.00 | 5.24 |

Note: The proportion of the changing area in the principal land-use pattern area is known as the converted land-use percent change.

### 3.3. SWAT Model Calibration and Validation at Subbasins of the SRYR

Simulated monthly runoff at Tangnaihai, Maqu and Jimai showed positive correlation against observed runoff during calibration and validation periods (Figure 7). For the certain runoff values throughout the peaks, the results remained underestimated or overestimated. At annual and monthly timescales, there was a good correlation between simulated and observed runoff at Tangnaihai, followed by the Maqu and Jimai subbasins (Table 8). Importantly, the PBIAS values for Tangnahai, Maqu and Jimai were 0.1%, −2.6%, and −5.1% during calibration and 3.1%, 2.0%, and 5.0% during validation, respectively. Overall, the SWAT model was more accurate during the validation period than during calibration, and performed well at the monthly timescale as compared to the annual timescale. The coefficient of determination ($R^2$) and Nash Sutcliffe Efficiency (NSE) values at monthly timescales ranged from 0.71 to 0.8 for the calibration period and 0.76 to 0.86 for the validation period.

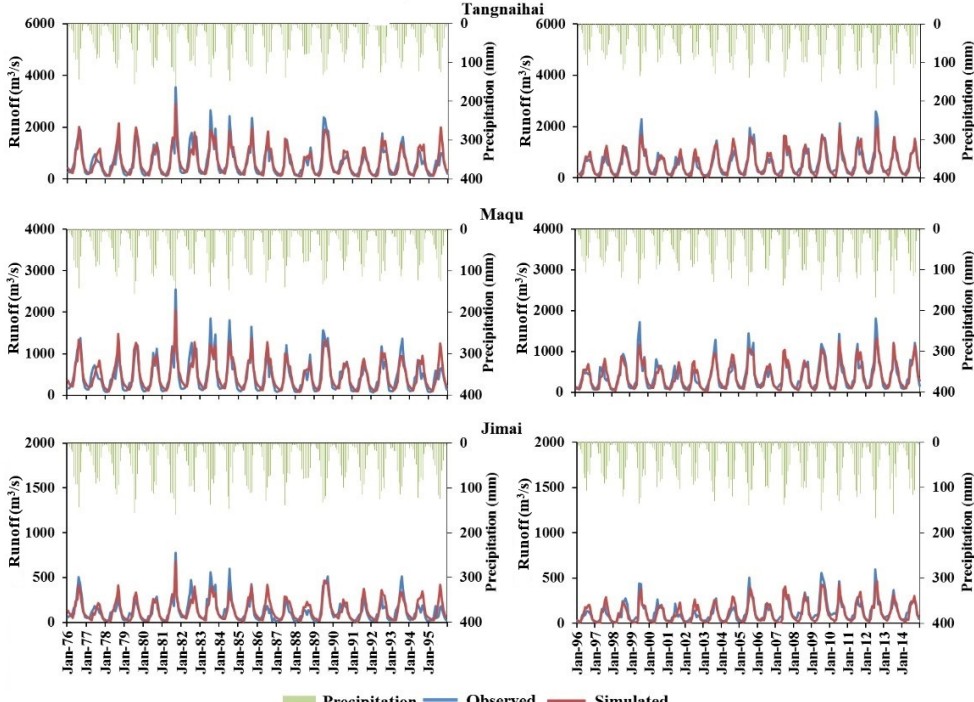

**Figure 7.** Comparison of monthly runoff during the calibration period (**left column**) and validation period (**right column**), based on land-use maps of the years 1990 and 2010, respectively.

**Table 8.** Performance evaluation indices of the SWAT model in the subbasins of the SRYR.

| Period | Basin | Monthly | | Yearly | | PBAIS |
|---|---|---|---|---|---|---|
| | | $R^2$ | NSE | $R^2$ | NSE | |
| Calibration (1976–1995) | Tangnaihai | 0.83 | 0.83 | 0.78 | 0.73 | 0.10 |
| | Maqu | 0.83 | 0.82 | 0.82 | 0.77 | −2.60 |
| | Jimai | 0.72 | 0.71 | 0.64 | 0.57 | −5.10 |
| Validation (1996–2014) | Tangnaihai | 0.86 | 0.85 | 0.82 | 0.75 | 3.10 |
| | Maqu | 0.85 | 0.84 | 0.82 | 0.76 | 2.00 |
| | Jimai | 0.77 | 0.76 | 0.79 | 0.69 | 5.00 |

Since the performance of the SWAT model could not achieve a very high accuracy at annual timescales, the results can be considered satisfactory in the Jimai subbasin. According to the classification of Moriasi et al. [52], the results of the SWAT model ranged from the satisfactory to very good performance classes. These findings demonstrated the applicability of the SWAT model simulations to determine the effects of climate change and LUC in the SRYR.

*3.4. Impacts of Climate Change and LUC on Runoff and Evapotranspiration at Annual Timescale*

In this section, simulated results were adopted instead of the observed data to compare the hydrological results in subbasins of the SRYR. Table 9 shows the simulated results of evapotranspiration and runoff by SWAT under four scenarios, as presented in Section 2.2.3.

**Table 9.** Simulated average annual runoff and evapotranspiration (mm) under different climate and land-use scenarios.

| Scenario | Climate | LUC | P | Runoff | ET | | Runoff Change | | ET Change | |
|---|---|---|---|---|---|---|---|---|---|---|
| | | | mm | mm | mm | | mm | % | mm | % |
| | | | | | Tangnaihai | | | | | |
| S1 | 1976–1995 | 1990 | 507.0 | 175.8 | 325.8 | | | | | |
| S2 | 1976–1995 | 2010 | 507.2 | 178.0 | 326.6 | $\Delta Q_C$ | −22.4 | 80.2 | 21.4 | 72.0 |
| S3 | 1996–2014 | 1990 | 509.6 | 161.1 | 339.6 | $\Delta Q_L$ | −5.5 | 19.8 | 8.3 | 28.0 |
| S4 | 1996–2014 | 2010 | 509.6 | 148.0 | 355.4 | $\Delta Q$ | −27.9 | 100.0 | 29.6 | 100.0 |
| | | | | | Maqu | | | | | |
| S1 | 1976–1995 | 1990 | 508.2 | 177.4 | 328.8 | | | | | |
| S2 | 1976–1995 | 2010 | 508.4 | 181.7 | 327.9 | $\Delta Q_C$ | −27.3 | 84.9 | 26.1 | 76.6 |
| S3 | 1996–2014 | 1990 | 511.5 | 159.2 | 346.0 | $\Delta Q_L$ | −4.9 | 15.1 | 8.0 | 23.4 |
| S4 | 1996–2014 | 2010 | 511.6 | 145.3 | 362.9 | $\Delta Q$ | −32.2 | 100.0 | 34.1 | 100.0 |
| | | | | | Jimai | | | | | |
| S1 | 1976–1995 | 1990 | 411.4 | 96.3 | 300.6 | | | | | |
| S2 | 1976–1995 | 2010 | 411.5 | 102.3 | 301.1 | $\Delta Q_C$ | −11.1 | 82.6 | 30.1 | 78.7 |
| S3 | 1996–2014 | 1990 | 428.8 | 93.5 | 323.1 | $\Delta Q_L$ | −2.3 | 17.4 | 8.1 | 21.3 |
| S4 | 1996–2014 | 2010 | 428.7 | 82.8 | 338.8 | $\Delta Q$ | −13.5 | 100.0 | 38.2 | 100.0 |

Note: For both runoff change and ET change, the total change $\Delta Q = (S4 − S1)$. For runoff change, $\Delta Q_C$ = average of $(S3 − S1) + (S4 − S2)$, and $\Delta Q_L$ = average of $(S2 − S1) + (S4 − S3)$. Percent % $\Delta Q_C = \Delta Q_C / \Delta Q \times 100$. Percent % $\Delta Q_L = \Delta Q_L / \Delta Q \times 100$.

3.4.1. Tangnaihai Subbasin

The simulated runoff declined by 27.9 mm (15.9%) and ET increased by 29.6 mm (9.1%) in the S4 scenario compared to the S1 scenario with respect to their totals. Thus, the combined influence of CC and LUC was noticeable from 1990 to 2010. The results suggested that, due to change in land-use, runoff decreased by 5.5 mm and ET increased by 8.3 mm, which, in turn, accounted for 19.8% and 28%, respectively, of the change in total runoff and ET. Climate variations also appeared to decrease the runoff and increase the ET by 22.4 mm and 21.4 mm, which comprised about 80.2% and 72.0% of the total changes,

respectively. Resultantly, CC was the predominant factor that affected the hydrological processes, while the influence of land-use changes was smaller.

### 3.4.2. Maqu Subbasin

The simulated runoff appeared to decrease, and ET appeared to increase by 32.2 mm and 34.1 mm, accounting for 18% and 10.4% of the totals. CC and LUC contributed to 27.3 mm (84.9%) and 4.9 mm (15.1%) of the total decline in runoff, respectively. CC and LUC contributed to 26.1 mm (76.6%) and 8.0 mm (23.4%) of the total increase in ET, respectively.

### 3.4.3. Jimai Subbasin

In the Jimai subbasin, the simulated runoff decreased by 13.5 mm and ET increased by 38.2 mm, accounting for 14% and 12.7% of the totals. In response to CC and LUC, runoff reduced by 11.1 mm and 2.3 mm, respectively, and ET increased by 30.1 mm and 8.1 mm. CC effected the ET in this subbasin more than the others because most of the subbasin area lies at a higher altitude.

### 3.5. Impacts of Climate Change and Land-Use Changes on Runoff and Evapotranspiration at the Monthly Timescale

Figure 8 elaborated the intra-annual runoff and ET variations caused by CC and LUC in all subbasins. A reasonable variation in runoff and ET affected by climate change became evident in different months, where the combined effect of climate parameters appeared. Since the air temperature and precipitation changes occurred over the whole year, changes in runoff were not correlated with precipitation (Table 4). Nevertheless, precipitation changes controlled the runoff from March to September. The variations in ET were correlated with runoff in May and June for Tangnaihai and in May for the Maqu and Jimai subbasins, respectively.

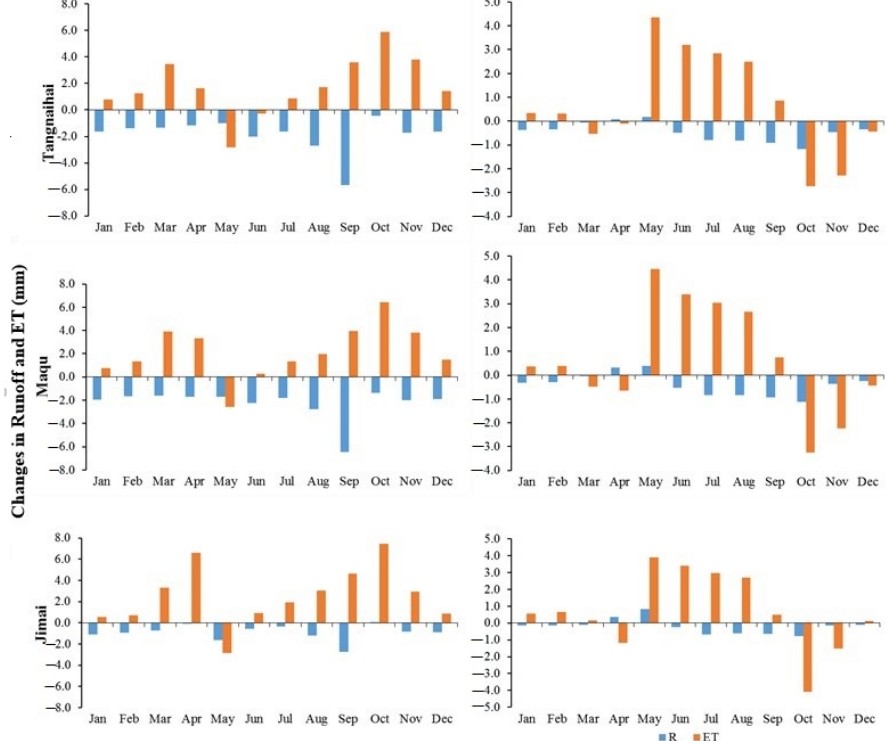

**Figure 8.** Monthly runoff (R) and evapotranspiration (ET) changes as a result of land-use changes (**right column**) and climate change (**left column**).

Figure 8 (right column) showed the effects of LUC on runoff and ET at the monthly scale in all subbasins. The graph clearly presented ET peaks in May and troughs in October

and November. The observed multiple peaks relate to the changes in vegetation. As per the analysis, the land-use change transformation of vegetation coverage from low to high can lead to an increase in ET due to the intercepted evaporation and vegetation transpiration.

However, the runoff decreased from April to October, which may be attributed to an increase in vegetation growth in the same period. This decrease in runoff has the potential to increase regulation capability and canopy interception, which, in turn, may have caused the changes to be stored in the root zone. It is interesting to know that, from November to March, the runoff increased as a consequence of the increased base flow driven by higher soil storage and, in April and May, runoff increased as a result of early snow-melting. The effect caused by the LUC was demonstrated to be relatively low, and varied between −3 and 5 mm for the Tangnaihai and Maqu subbasins and between −4 and 4 mm for the Jimai subbasin.

## 4. Discussions

For SWAT, the sensitive parameters were primarily adopted from the previous studies conducted in the whole Yellow River basin and the SRYR, as mentioned earlier (reference). The sensitive parameters can be characterized into groups of soil, groundwater, basin and HRU. However, the optimal values of the sensitive parameters differ, which could be explained as the difference in the climate data period that was used, as well as differences in the sources of other physical parameters, such as land-use or soil data. For example, the optimal value of the soil evaporation compensation factor (ESCO) obtained by [27] was 0.8, which was slightly than that of this study, where its value was optimized at 0.95. ESCO accounts for the effects caused by cracking, crusting and capillary action, allowing fpr the user to fix the distribution of soil evaporation demand. The higher values of ESCO (0.95) may be related to the fact that soil evaporation is not significant in the evaporation process, which may be due to the high vegetation coverage in the SRYR. CN2 is a highly sensitive parameter in all the watersheds, reflecting characteristics such as soil type, soil moisture, and management practices in the land. This affects the surface runoff which, in turn, affects the total runoff. GW_DELAY is the parameter dealing with the geology of the region, i.e., it indicates the delay in time between the water exiting the soil profile and moving to the shallow aquifer. The higher value of GW_DELAY (105.72) can be elucidated by the lower water content in the vadose zone due to the increasing evapotranspiration that plays an important role in the hydrological processes in the region [9]. Usually, mostly hydrological models are set under the assumptions that the influence of climatic conditions is comparatively more pronounced than the change in calibrated parameters, which represents the biasness of the hydrological model [53].

The transition matrix of land-use change presented in Table 6 indicates that the more pronounced changes are the reduction in sparse grassland and augmentations in grassland, unused land and woodland; however, these LUC transitions vary among the subbasins of the SRYR. The decrease in sparse grassland can principally be attributed to the transition into woodland, unused land and grassland. In the Jimai subbasin (a higher-altitude area), the conversion of sparse grassland to unused land may lead to a reduced interception and water retention, causing a reduced decrease in runoff, which is in agreement with [6,14]. The degradation of sparse grassland in the Jimai subbasin may be due to the more pronounced, warmer climate, as reported by [2,7,9,16], and lower number of rainy days (>0.1 mm per day) [39]. However, in the Maqu subbasin, the major conversion in land-use is to an increase in grassland and woodland (Figure 6). This is due to the reduced decrease in the frequency of precipitation events and increased agricultural activities in low-lying areas of the Maqu and Tangnaihai subbasin, as reported by [39]. Overall, in the Maqu subbasin, the conversion to vegetation is greater as compared to the others, which could ultimately decrease the water supply. Yang et al. [14] stated that this increase in vegetation leads to an increase in transpiration and canopy interception, which may affect the response to runoff. The results of the degraded land in the Jimai subbasin are consistent with the studies of [27,28,54], where they reported on the significant degradation in the alpine ecosystem in

the the last 30 years, which seriously affected the regional water storage capacity [55]. The increase in grassland in the Maqu and Tangnaihai subbasin due to the Three Rivers Source Area Ecological Protection Project greatly enhanced the vegetation coverage condition and increased the water storage capacity of the region [11].

The impacts on hydrological processes (runoff and evapotranspiration) can be synthesized by considering the integrated approach, consisting of the hydrological modelling and statistical framework proposed by [14], since the total contribution of CC and LUC approached 100%. Overall, the reduction in runoff and enhanced evapotranspiration climate remained dominant in all three subbasins compared to land-use; however, the magnitude of impacts due to climate and land-use varies among subbasins, which may attribute to the climate and land-use variability of a particular subbasin. The pattern of reductions in runoff of 11.1 mm, 27.3 mm and 22.4 mm in Jimai, Maqu and Tangnaihai, respectively, was in accordance with the conclusions of Meng et al. [9], where 3.2 mm, 9.2 mm and 6.0 mm reductions in runoff per decade were concluded for these subbasins, respectively. The increment in evapotranspiration in subbasins was also in agreement with [9]. A detailed analysis of recent studies that explored the effects of CC and LUC on hydrology revealed that the contributions of the two components discussed in this article may vary in different study areas [18,19,56]. According to Ma et al. [57], LUC can have a significant influence, altering the streamflows of subtropical watersheds, as the LUC was primarily driven by anthropogenic activities. The effects of LUC on streamflows were more pronounced than those of CC, as explored by Li et al. [58] and Saifullah et al. [59]. Due to the regional diversity of CC and LUC, the varied impacts of the two components in the various research regions is reflected accordingly. The decline in runoff found in our study, caused by CC in subbasins of the SRYR, which was significantly larger than the decline caused by LUC, presented CC as the dominant factor affecting the hydrological processes. This result is in line with the results of Cuo et al. [60], who found that the hydrological changes above Tangnaihai are mostly caused by CC, and the consequences of LUC and reservoir release become significant below Tangnaihai. These results were also consistent with those of Pan et al. [11], who found that water supply noticeably decreased from 1980 to 1985, and CC remained the dominant factor decreasing the water supply in the SRYR. However, from 2000 to 2005, due to an increase in precipitation, CC played a positive role in increasing the water supply [11].

The reduction in runoff and increase in evapotranspiration in the Jimai subbasin may be attributed to the warmer climate and reduced canopy interception. However, the magnitude of change in both runoff and evapotranspiration was greater in the Maqu and Tangnaihai subbasins because of the increased temperature and interclass land-use changes. These results are similar to those of [9] and in contradiction with [14], where the response of land-use changes suggested an incremental runoff. This could be more or less than that which was carried using the simulated results of a single period, from 1960 to 1979, or in different periods, indicating that LUC was not apparent compared to climate change in a different period. Basins with varied climates, rainfall–runoff processes, and patterns of land cover often cannot be studied with such an approach. The results suggest that better management is necessary for the SRYR, where few human activities have occurred [5,61,62].

This study also emphasizes the vital demand for an integrated model that incorporates both CC and LUC impacts to appropriately analyze the hydrological processes. In contrast to the conventional OFAT approach, a novel method that combines statistical analysis with hydrological modelling may consider the interaction between CC and LUC's impacts on hydrological processes during the entire study period.

## 5. Conclusions

Understanding the relationship between variations in hydrological processes has become more crucial as China's water resources become scarcer. It has been established that the climatic conditions have the greatest impact on these activities. Additionally, it was believed that LUC had a significant impact on how the hydrological processes occurred.

Recently, it has been suggested that one of the causes of CC and the primary contributor to the LUC is anthropogenic activity. Climate change undoubtedly can alter the flora in arid and semi-arid areas, which will then have an impact on the hydrology of the area. Therefore, LUC as well as CC has an impact on the area streamflows. In this study, the SWAT model was successfully applied in three subbasins, Tangnaihai, Maqu, and Jimai, of the SRYR. Later, to distinguish the impacts of CC and the LUC on runoff and ET, SWAT model results were obtained for four scenarios (based on two land-use maps and two CC periods). The important conclusions are as follows:

- Broadly, the results characterized the SRYR as warmer, wetter, and showing a greater decrease in water availability during the period 1974–2014. Average annual temperature experienced a significant increasing trend, with a warming rate of 0.4 OC per decade. The average annual precipitation presented a slight increase of about 3.64 mm per decade. Runoff showed decreasing trend of about 36.4 m$^3$/s per decade in the region.
- Analysis of the land-use changes revealed an increase in grassland, unused land and woodland, while a decrease was found in sparse grassland. Land-use transformation showed an increase in grassland, which accounted for 5.2% of the entire area.
- The SWAT model simulations showed that the integrated impacts of climate and land-use changes caused a decrease in runoff of 27.9 mm, 32.2 mm and 13.5 mm in the Tangnaighai, Maqu and Jimai subbasins; that is, 15.9%, 18% and 14% of the total runoff, respectively. The results establish that climate change has more pronounced impacts than land-use changes, causing decreases in runoff measured at 80.2%, 84.9% and 82.6% of the total runoff changes at the Tangnaighai, Maqu and Jimai subbasins, respectively. The land-use changes also showed a decrement of 19.8%, 15.1% and 17.4% of the total runoff changes.
- In contrast to the runoff, ET increased by 29.6 mm, 34.1 mm and 38.2 mm in the Tangnaighai, Maqu and Jimai subbasins, respectively, where climate change played a more important role (>70%) than land-use changes (<30%) in increasing the ET.
- The findings also demonstrated that most of the reduction in runoff occurred in the Maqu subbasin and there was a greater increase in ET in the Jimai subbasin, which may be attributed to the increase in vegetation and unused land in the respective subbasins.

Therefore, the CC and LUC had different impacts on runoff and ET depending on the changes in these variables in a particular basin. As an important strategic result, this study delivers possible pathways that could be considered by numerous policy-makers when designing adaptive measures for CC and planning sustainable development policies for the ecological systems of the SRYR.

**Author Contributions:** Conceptualization, J.W.; data curation, M.I.; formal analysis, M.I. and M.U.M.; supervision, J.W.; validation, M.A.; visualization, M.M.; writing—original draft, M.I.; writing—review and editing, J.W., M.M., M.A. and M.U.M. All authors have read and agreed to the published version of the manuscript.

**Funding:** The Sichuan Provincial Science and Technology Planning Project (2021YJ0025), the Chinese National Natural Science Foundation (41971308) and the Scientific Research Foundation of CUIT (KYTZ201821) provided financial support for the study.

**Institutional Review Board Statement:** Not applicable.

**Informed Consent Statement:** Not applicable.

**Data Availability Statement:** The data used in this study (climatic, hydrological and land-use) were requested from the relavent department, as mentioned in data section of this manuscript. However, the SWAT model, DEM, and soil raster data are freely available (details given in the data section).

**Acknowledgments:** We thank the Yellow River Conservancy Commission and the China Meteorological Administration (CMA) for supplying the hydro-meteorological data that were needed to undertake this study.

**Conflicts of Interest:** In this research, the authors report no conflict of interest.

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
