# Peer review of "Impacts of Climate and Land-Use Changes on Hydrological Processes of the Source Region of Yellow River, China"

_sustainability, doi:10.3390/su142214908_

Round 1

Reviewer 1 Report

Comments have been attached 

Author Response

Response to the comments of Reviewer Report 1

Comments: In this study, authors quantified the impacts of climate change and land use change on the streamflow and evapotranspiration. They mainly used MK test and SWAT model to complete their objectives. They conclude that climate change is the major reason in streamflow decrease and evapotranspiration increase. Recently, this topic is getting intension by researchers in the field of climate change and hydrology. However, due to poor language, lack of proper writing, putting some wrong citations, some flaws in methodology (described below), I cannot recommend this manuscript for publication in this reputed journal.

Response: We are very much thankful for your positive comments, and your suggestions and comments will be useful in improving our manuscript. We took into account these comments and suggestions, and improved the writing, corrected wrong citations, and modified the manuscript accordingly. We have highlighted the modifications with “tracked changes on” in the manuscript. Number wise answers to the comments/suggestions are as follows. Hope this revision could meet the requirements of the Journal.

General comments:

Q01: Language: Language is so poor, which needs extensive editing. Many time, it is not even understandable.

A01: Thanks for the observation. We have reviewed the language extensively and elucidated where required and hope it is now a much better for the revised manuscript.

Q02: Abstract: This section need rewrite because it is not completely understandable. There are some grammatical errors.

A02: Thanks for your comments. The abstract act as summary of this research. We have revised it both grammatically and structure-wise and come up with a most refined version of abstract.

Q03: Introduction section is so poorly constructed. There is no flow from start to end. A proper background, previous literature, and problem statement need extensive improvement. Authors need to highlight the novelty of the work, which is completely missing. Authors also need to avoid giving the wrong citation such as Zhang et al. (18) and Genxu et al. (33).

A03: Thanks for your guidelines. The manuscript’s structure of the introduction section has been revised and improved . Extensive efforts have been made to make it very clear and to maintain the flow along novelty by adding a complete paragraph highlighting the gaps of previous studies. As of the mentioned citations, these are just misplacement of citations with the different references of the same authors and we have replaced them with the right citations in the revised manuscript. Hope our endeavor will fulfill the requirement.

Q04: Method: For Mann-kendall test is necessary to deal with the serial correlation, which can mislead the trend results. Provide proper SWAT model description and also describe clearly why do you need and what is the advantage of Method 2 (CC and LUC) over Method 1

A04: Thanks for your guideline. Mk test requires the data should be free from serial correlation (Von Storch 1999) to preserve the actual trend. However proposed procedure for removal of the serial correlation effect i.e pre-whitening technique compromise the originality of time series and removes a portion of a trend (Douglas et al. 2000; Yue et al. 2002). We have confirmed the pattern among Precipitation and Runoff  from Figure 3 and Figure 5a of Yuan et al. 2015 and similar behavior of spikes is found.  About serial correlation in temperature data that we have already addressed in previous study Iqbal et al. 2018a for temperature data by using trend free pre-whitening technique (TFPW). About Runoff data, Hu et al. 2011 has explained that none of the data series for detecting trend has significant serial correlation at a 5% level.

Interestingly, parametric test (e.g. linear regression method) which is considered as more powerful over non-parmetric (e.g. MK test) but requires data to be distributed normally. By applying linear regression method, trend was found to be positive for precipitation data in the previous study (Iqbal et al. 2018b).. Meng et al. 2016 has also showed slightly increasing trend in precipitation and decreasing trend of runoff. Similar result we found in this study also.

Now in revised manuscript we have referred these studies about the quality and standard approaches for data analysis in the methodology section. 

About Swat model description and its need, the description is revised and augmented highlighting the suitability in prospective of the SRYR. Also the method 2 advantage is placed in the section of 2.2.3 of methodology in  the revised manuscript.

Hope the this answer will satisfy your concerns. The referred studies list is as below;

Von Storch, H., 1999. Misuses of Statistical Analysis in Climate Research. In: Analysis of Climate Variability. Springer, pp. 11–26.

Douglas, E.M.; Vogel, R.M.; Kroll, C.N. Trends in floods and low flows in the United States: Impact of spatial correlation. J. Hydrol. 2000, 240, 90–105, doi:10.1016/S0022-1694(00)00336-X.

Yue, S.; Pilon, P.; Cavadias, G. Power of the Mann-Kendall and Spearman’s rho tests for detecting monotonic trends in hydrological series. J. Hydrol. 2002, 259, 254–271, doi:10.1016/S0022-1694(01)00594-7.

Yuan, F.; Berndtsson, R.; Zhang, L.; Uvo, C.B.; Hao, Z.; Wang, X.; Yasuda, H. Hydro Climatic Trend and Periodicity for the Source Region of the Yellow River. J. Hydrol. Eng. 2015, 20, doi:10.1061/(asce)he.1943-5584.0001182

Iqbal, M.; Wen, J.; Wang, X.; Lan, Y.; Tian, H.; Anjum, M.N.; Adnan, M. Assessment of Air Temperature Trends in the Source Region of Yellow River and Its Sub-Basins, China. Asia-Pacific J. Atmos. Sci. 2018a, 54, 111–123, doi:10.1007/s13143-017-0064-x.

Iqbal, M.; Wen, J.; Wang, S.; Tian, H.; Adnan, M. Variations of precipitation characteristics during the period 1960–2014 in the Source Region of the Yellow River, China. J. Arid Land 2018b, 10, 388–401, doi:10.1007/s40333-018-0008-z.

Hu, Y.; Maskey, S.; Uhlenbrook, S.; Zhao, H. Streamflow trends and climate linkages in the source region of the Yellow River, China. Hydrol. Process. 2011, 25, 3399–3411, doi:10.1002/hyp.8069.

Meng, F.; Su, F.; Yang, D.; Tong, K.; Hao, Z. Impacts of recent climate change on the hydrology in the source region of the Yellow River basin. J. Hydrol. Reg. Stud. 2016, 6, 66–81, doi:10.1016/j.ejrh.2016.03.003.

Q05: Results: Please revise the results section and remove the redundancy and do not describe everything given in Figures and Tables. Just highlight the interesting results.

A05: Thank you for the comments. We have modified the results section and removed redundancy as per suggestions. We hope the modification can meet your requirement.

Q06: Discussion section: Authors can discuss anything in this section and it can be as long as they can but this section should be based on the main results (such as impacts of CC and LUC on runoff and ET), according to the objective and thrust of the study. No need to discuss irrelevant studies, for example what is the resemblance between this study and Chung et al. (23)? Be concise and concentrate on the interesting and unique results of the study.

A06: Thanks for your guidelines. In revised manuscript, we have deleted the discussion of most common results. However the improved discussion about the results of landuse changes are kept, because the changes of runoff or ET due to landuse change could be attributed. Also avoided to discuss the irrelevant studies and citations.

Q07: Conclusion section: conclusion should be conclusion not discussion as author did from 625-640. Authors just need to describe conclusion. For example, "Climate change is the dominant factor affecting streamflow and ET in the region." no need to justify here that it is happened because of this and that. Just add 3-5 main conclusions better in the form of bullets.

A07: Thank for your comments and guidelines. There can be different ways to write this section. Following the paragraph or in bullets form. Now we have updated this section with five main conclusions and wrote those in bullets form. Also, we have removed the justification in revised manuscript because such things we have discussed already in discussion section.

Specific Comments:

Q01: Line 15: "the Source Region of the Yellow River". "the" article missing be Yellow River

A01: Thanks for your carefully. We have reviewed the line grammatically and added articles where required. We hope this will meet your requirement.

Q2: Line 17: Land-use changes or Land use changes, be consistent

A02: Thank you for comment. We have improved the MS and tried to be consistent with the terms used in the whole manuscript. Please see the revised manuscript.

Q03: Line 19: Spatially, not spa-tially. Check similar kind of word throughout the MS

A03: Sorry for appearing this mistakes We have reviewed the MS grammatically and edited it where required. Hopefully, it meet your requirement.

Q04: Line 20-23: revise please

A04: Thanks for the comment. We have revised the sentence in the revised manuscript and hope the description is in much better form now.

Q05: Line 21: decrease not decrease

A05: Thanks for the observations. We have reviewed it and elucidated where required. We hope, this will fulfil your requirement.

Q06: Line 23: The changes in runoff and ET not "the changes of runoff and ET"

A06: Thanks. We have corrected it in the revised manuscript as your suggested.

Q07: Line 27: Subbasin is established word in Hydrology, so use subbasin instead sub-basin

A07: Thanks for pointing out this error. We have changed the all word “sub-basin” to “subbasin” in the revised manuscript.

Q08: Line 23-28: there are some grammatical errors in the sentences. revise it

A08: Thanks for the noticing. We have gone through the whole manuscript grammatically and tried to furnish this issue. The editing was made where required. However, the said sentence is reproduced here below. We hope this will meet your requirement.

“The changes in runoff and ET caused by LUC seemed to be adequate by comparison and presented 15.1-19.8% decline in runoff and 21.3-28% increase in ET relative to the totals. Overall, climate change has more influence on hydrological processes in all subbasins of the SRYR as compared to LUC. It is therefore constructed that response to change hydrological process in a subbasin may be attributed to change in individual climate parameter and landuse type.”

Q09: Line 33-44: Rewrite please

A09: Thank you for your guideline. We have rewritten the line and hope this will meet the standard of research.

Q10: Line 46: What does this mean "In past, studies have been undertaken to examine climate change effects on hydrological components, in which its role has been identified"

A11: Thanks for your comment. The purpose of this was to mention about the advancements in this research. Firstly we explain about the studies those focus only on climate change impacts on hydrological components and in subsequent sentence we mentioned about the studies on climate change and landuse change impacts on hydrological components.

Now for easiness, we have revised it in the revised manuscript and also provided here below. We hope this address your concern.

"In past, studies have been conducted to examine effects of climate change on hydrological components, in which its parameters (i.e. temperature and precipitation) role has been identified".

Q12: Line 51: what does author mean "separate impacts of climate change and LUC"

A12: Sorry for this confusion. we want to emphsis that “individual impact of climate change and landuse change”. We hope this will satisfy your requirements.

Q13: Line 55: "climate change was more influenced than LUC" not clear In that study, Daofeng et al did not separate the impacts of climate and Land use on streamflow

A13: Thanks for the observations. The said study reference is provided here below. The author’s work is related to impacts of climate and landuse on streamflow by developing different scenarios of land cover change and climate change or different combinations of temperature and precipitation.

However, what we mentioned in the manuscript was “Daofeng et al. [25] presented only the impacts of streamflows in Yellow River’s upper reaches where climate change was more influenced than LUC”. Here “only the impacts of streamflow” becomes the reason of confusion.

Now this quotation has been changed in revised manuscript and reproduced here below.

“Daofeng et al. [26] presented impacts of CC and LUC on streamflows in the Yellow River’s upper reaches where climate change was more influenced than LUC.”

Daofeng, L.; Ying, T.; Changming, L.; Fanghua, H. Impact of land-cover and climate changes on runoff of the source regions of 712 the Yellow River. J. Geogr. Sci. 2004, 14, 330–338, doi:10.1007/bf02837414.

Q14: Line57: Grammatically incorrect

A14: Thans. We have revised thegramma in the line. We hope this will meet your requirement.

Q15: Line 67: are the authors sure about this statement "The major ecosystem covered by alpine meadows has been degraded remarkably in past three decades" because Chinese GOVT. have launched different project in the Sanjiangyan to restore the habitats such as Ecological Protection and Restoration Program

A15: Thanks you for your questions. First it should be clear that the stud area is the Yellow River, not the Sanjiangyuan, it is part of the Sanjiangyuan. Sorry that we didn't clarify this issue. In fact, before the Chinese government implemented the habitats such as Ecological Protection and Restoration Program, the ecosystem in the source area of the Yellow River was degraded, while after the Program, the ecosystem in the source area of the Yellow River gradually restored gradually. See details in the revised manuscript.

Yes, according to studies of Salhab et al. (2010) and Wen et al. (2013), the major ecosystem in the region has been degraded. Chinese Government project, Ecological Protection and Restoration Program for Three-River Headwaters (TRH, the headwaters of the Yangtze, Yellow, and Lantsang rivers) region initiated in 2005. In the result of EPRP implementation, vegetation cover-Normalized Difference Vegetation Index (NDVI), increased by 11.2% and grassland yield has also been increased which reduce the grazing pressure in the region (Zhang et al. 2017). These increments in vegetation cover and grassland yield vary with respect to space and prominent in eastern side which is tallying with our study. According to Figure 2 (landuse map of the year 2010), it can be seen that grassland has increased as compared to 1990 landuse map. It means that we have modelled the results using 1990 and 2010 landuse maps of before and after EPRP.

Now we have added vegetation improvement by EPRP next to said Line (67) in the revised manuscript. Hope this will satisfy your requirement. The refereed researches are listed below:

Salhab J, Wang J, Anjum SA, Chen Y (2010) Assessment of the grassland degradation in the southeastern part of the source region of the Yellow River from 1994 to 2001. Journal of Food Agriculture and Environment 8(3–4): 1367–1372.

Wen L, Dong S, Li Y, Li X, Shi J, Wang Y, et al. (2013) Effect of Degradation Intensity on Grassland Ecosystem Services in the Alpine Region of Qinghai-Tibetan Plateau, China. PLOS ONE, 8(3): e58432. doi: 10.1371/journal.pone.0058432 PMID: 23469278

Zhang, L.; Fan, J.; Zhou, D.; Zhang, H. Ecological Protection and Restoration Program Reduced Grazing Pressure in the Three-River Headwaters Region, China. Rangel. Ecol. Manag. 2017, 70, 540–548, doi:10.1016/j.rama.2017.05.001.

Q16: Line 73-76: "Author says that according to Zhang et al., climate change caused more impact on streamflow" But Zhang's study is not about computing the effect of CC and LULC on streamflow. They used wavelet transform to find links between streamflow and temperature and precipitation.

A16: Thanks for your comments. This was happened due to the same last name while adding the citation. We have replaced the citation with correct one. Previous citation (18) and corrected one (34) in revised manuscript are given below. Except this, authors placed new citations in the revised manuscript which is related to LUC and CC effects on hydrological processes. Hope this will furnish your comments.

18.Zhang, J.; Li, G.; Liang, S. The response of river discharge to climate fluctuations in the source region of the Yellow River. Environ. Earth Sci. 2012, 66, 1505–1512, doi:10.1007/s12665-011-1390-4.

34.Zhang, Y.; Zhang, S.; Zhai, X.; Xia, J. Runoff variation and its response to climate change in the Three Rivers Source Region. J. Geogr. Sci. 2012, 22, 781–794, doi:10.1007/s11442-012-0963-9.

Q17: Line 74: "Genxu et al. [33] concluded that degradation of ecosystem due to anthropogenic activities is a primary cause for its reduction", But Genxu's article is about "dynamics of soil water content and water movement in the active layer of alpine meadow soils were monitored at sites located in the permafrost region of the Qinghai-Tibetan Plateau, China for four years to examine the synergistic effects of freeze-thaw cycles and levels of vegetation cover" Can you justify this wrong citation?

A17: Thanks for noticing the details of the author’s work. Autually when we go through the literature, we read the quoted citation also for providing the evidences with more citations. Since the researches of Genxu et al (2009)/Wang et al (2009) are as explained in comment, however Genxu et al (2009)/Wang et al (2009) work quoted by Pan et al (2005) is as “Wang et al [31] believe that ecosystem degradation due to human activity is the primary reason for the reduced runoff as the ecosystem services were degraded.” References as discussed above are given below:

Now we have deleted this citation and added other related citations in revised manuscript. Actually authors present the previous work where in some places climate change has more influence on hydrological processes and in some places LUC has more influence.  

  1. Pan, T.; Wu, S.; Liu, Y. Relative contributions of land use and climate change to water supply variations over yellow river source area in Tibetan Plateau during the past three decades. PLoS One 2015, 10, 1–19, doi:10.1371/journal.pone.0123793.
  2. Genxu, W.; Shengnan, L.; Hongchang, H.; Yuanshou, L. Water regime shifts in the active soil layer of the Qinghai-Tibet Plateau permafrost region, under different levels of vegetation. Geoderma 2009, 149, 280–289, doi:10.1016/j.geoderma.2008.12.008.
  3. Wang, G., Li, Y., Wang, Y., Shen, Y., 2007. Impacts of alpine ecosystem and climatic changes on surface runoff in the source region of Yangtze River. Journal of Glaciology and Geocryology 29 (2), 159–168.

Q18: Line 102: "Anyamaqin Mountain and drainage outlet" this cant be the reason of change in elevation

A18: Thanks for your comments. Authors supposed that higher elevation change exist in Tangnaihai subbasin, as compared to Jimai, and Maqu subbasins. This change is due to peaks of mountain and lowest elevation areas (e.g. outlet of the SRYR) exist in the Tangnaihai subbasin. We have revised this sentence in the revised manuscript. Hope the sentence is more understandable now.

Q19: Line 135: correct references as [2, 37], throughout the MS

A19: Thanks for the observation. We have removed the issue throughout the manuscript however the citation sequence is changed now because of newly added citations in the revised manuscript.

 Q20: Table 1: Under Source, provide the references of datasets

A20: Thanks. We have updated the table with required references in revised manuscript.

Q21: Table 2: In this table is require to include in the MS, as these information is available in Fig. 1? it looks like not necessary

A21: Thank you for the suggestion. We have deleted the Table 2 and accordingly the manuscript has revised. We hope this will meet your requirement.

Q22: Line 169: Add reference instead of link    

A22: Thanks . We have removed the link as it has been mentioned in Table 1of the revised manuscript. Also we have place the reference as suggested.

Q23: Line 177; Gelic Leptosols and Mollic Leptosols. please describe little bit, these are not understandable to non-soil related readers

A23: Thanks for the guidelines. We have described the soils to be understandable for easiness of the readers. The material related to the supporting material is given below. We hope this will satisfy your concern.

“Gelic properties refer to soils having permafrost within 200 cm from the soil surface. Mollic lepotols refers to the soils which contains or immediately overlies with greater than or equal to 40.0% of calcium carbonate content.”

Q24: Line 188: correct citation

A24: Thanks for your comment. We have added the correct reference along with the link and hope this will satisfy your concern

Q25: Line 207-208: Does SWAT model has only SCS method to calculate losses and precipitation access?

A25: Thanks for your observations. There are two methods in SWAT to compute surface runoff namely Green-Ampt infiltration method and SCS curve number method, and we have mentioned both of them in revised manuscript. Hope this will answer your requirement.

Q26: Line 230: is it heading?

A26: Thanks for the comments. No it is not, it is just the prologue of the coming paragraph as we have mentioned Model Setup and Calibration and Validation (line) – under the heading 2.2.2 Hydrological Modelling.

Q27: Line 246-248: Author did not describe how they use the Land Use during calibration and validation. I mean which parameters obtained from Land Use?

A27: Thanks for your question. The said lines are produced here for clarification on this comment.

“The SWAT model was jointly calibrated from 1976 to 1995 using landuse of the year 1990 and validated for the period of 1996 to 2014 with the landuse 2010 against the observed streamflows at Jimai, Maqu, and Tangnaihai hydrological stations”.

Since to meet the objective of research (i.e. LUC and CC impacts on ET and runoff), normally scientist adopt this procedure to segregate the impacts of LUC and CC using two climate period and two landuse maps or we can say with a statistical framework. Selection of Landuse can be done in such a way that map represents the landuse characteristics of that time period. (e.g. 1990 landuse map for 1976-1995 time period).

Now continuing the research in Hydrological model (SWAT model), firstly we calibrate the model (by using forcing meteorological data of first period with corresponding landuse map) with different types of parameter (e.g. soil, basin, curve number representing the landuse or others). Keeping the same parameters values we validate the model for second climate period and its corresponding landuse map. List of parameters is provided in Table 2 where fitted value of curve number representing the landuse is mentioned.

Similar procedure has been adopted by numerous researchers. Few of those are mentioned here for your kind consideration.

Ahmed, N.; Wang, G.; Booij, M.J.; Xiangyang, S.; Hussain, F.; Nabi, G. Separation of the Impact of Landuse/Landcover Change and Climate Change on Runoff in the Upstream Area of the Yangtze River, China. Water Resour. Manag. 2022, 36

Zhang, L.; Nan, Z.; Yu, W.; Zhao, Y.; Xu, Y. Comparison of baseline period choices for separating climate and land use/land cover change impacts on watershed hydrology using distributed hydrological models. Sci. Total Environ. 2018, 622–623, 1016–1028, doi:10.1016/j.scitotenv.2017.12.055.

Yang, L.; Feng, Q.; Yin, Z.; Deo, R.C.; Wen, X.; Si, J.; Li, C. Separation of the Climatic and Land Cover Impacts on the Flow Regime Changes in Two Watersheds of Northeastern Tibetan Plateau. Adv. Meteorol. 2017, 2017, doi:10.1155/2017/6310401.

Q28: Table 3: How realist are the parameters estimated in this study? compare with other studies and discuss

A28: Thanks for the observations. Parameters were sorted out initially from the studies conducted in the whole Yellow River Basin and the SRYR [5, 31, 51]. Then sensitivity analysis was performed. Calibrated parameters are placed in the table 2 with max., min and fitted values in the revised manuscript. This was also mentioned in the previous version of manuscript. In revised version we explain it more clearly in the subsection 2.2.2 of methodology. We have also discussed some important parameters in start of the discussion section. The referred studies list is given below.

  1. Xu, Z.X.; Zhao, F.F.; Li, J.Y. Response of streamflow to climate change in the headwater catchment of the Yellow River basin. Quat. Int. 2009, 208, 62–75, doi:10.1016/j.quaint.2008.09.001.
  2. Liu, L.; Liu, Z.; Ren, X.; Fischer, T.; Xu, Y. Hydrological impacts of climate change in the Yellow River Basin for the 21st century using hydrological model and statistical downscaling model. Quat. Int. 2011, 244, 211–220, doi:10.1016/j.quaint.2010.12.001.
  3. Zhang, Y.; Su, F.; Hao, Z.; Xu, C.; Yu, Z.; Wang, L.; Tong, K. Impact of projected climate change on the hydrology in the headwaters of the Yellow River basin. Hydrol. Process. 2015, 29, 4379–4397, doi:10.1002/hyp.10497.

Q29: Line 285: which hydrological processes?

A29: Thanks for the comment. By the hydrological process herein we mean “runoff and evapotranspiration”. We have mentioned these in the revised manuscript. Hope this will satisfy your requirement.

Q30: Line 284: How method 2 is different from Method 1 and what is the advantage on Second method on method 1?

A30: Thanks for the comment. In accordance with literature, an accurate identification of the individual impacts of CC and LUC on hydrological processes cannot be accomplished with conventional One Factor at a Time (OFAT)-method 1 because the overall contribution do not meet up to be100%. For an instance, Li et al. (2009) studied an agricultural catchment area on the Loess Plateau in China during the period 1981 to 2000 and identified the impacts of climate change and LUCC on decreasing river flow amounting to about 95.8% and 9.6%, respectively. This indicate that other factors were involved.

A basic flaw of the conventional OFAT method is its inherent assumption that in the course of evaluating the influence of a given factor (e.g., climate change) on hydrological processes, the other factor (e.g., land use/cover) does not change for the entire period. Thence, separating the influence of climate change on hydrological processes warrants the inputting of only status of land cover for the baseline as a factor applicable to the entire study period that has considered in modified OFAT (method 2).

Hope the response will meet your requirement.

Li, Z.; Liu, W. zhao; Zhang, X. chang; Zheng, F. li Impacts of land use change and climate variability on hydrology in an agricultural catchment on the Loess Plateau of China. J. Hydrol. 2009, 377, 35–42, doi:10.1016/j.jhydrol.2009.08.007.

Q31: Figure 4: 10 years trend no meaning because for climate studies 30 years period is recommended by IPCC to see trend. The overall trend from 1974-2014 make sense but from 1974-1983. For example, in temperature graph, temperature is decreased during first decade but temperature in the region is continuously increasing since 1951. So better to use over trend from 1974-2014 and use the decadal average

A31: Thanks for the comment. Researchers had conducted decadal analysis considering either mean or trends to see the inter-decadal variability in data. For an instance, references of two studies considering decadal trends are provided here. Statistical tests require the sample size of 10 or more than 10 to detect the trend in data for Mann-Kendall test statistics S or Z, respectively. However, IPCC recommended the data sample size of 30 for climate studies.

Now we have compared the variability with mean as suggested in revised manuscript and kept the trends values for the whole period (1974-2014). Hope the answer will fulfill your requirement.

Ahmed, N.; Wang, G.; Booij, M.J.; Xiangyang, S.; Hussain, F.; Nabi, G. Separation of the Impact of Landuse/Landcover Change and Climate Change on Runoff in the Upstream Area of the Yangtze River, China. Water Resour. Manag. 2022, 36, 181–201, doi:10.1007/s11269-021-03021-z.

Wang, J.; Yan, Z.; Jones, P.D.; Xia, J. On “observation minus reanalysis” method: A view from multidecadal variability. J. Geophys. Res. Atmos. 2013, 118, 7450–7458, doi:10.1002/jgrd.50574.

Q32: Figure 4: How can be precipitation positive high spike but runoff negative high spike. this can be the affect of serial correlation. Please apply serial correlation before MK test and also check data and provide some positive reason

A32: Thanks for the observation. The part of this comment is also answered in general comments. Here we are reproducing that answer.

Mk test requires the data should be free from serial correlation (Von Storch 1999) to preserve the actual trend. However proposed procedure for removal of the serial correlation effect i.e pre-whitening technique compromise the originality of time series and removes a portion of a trend (Douglas et al. 2000; Yue et al. 2002).

Out of 40 spikes of runoff, only 4 spikes or runoff values are not following the spikes of precipitation for the years of 1985, 1991, 1995 and 2001. We have confirmed the pattern among Precipitation and Runoff of these years from Figure 3 and Figure 5a of Yuan et al. 2015 study and similar behavior is found.  About serial correlation, which we have already addressed in previous study Iqbal et al. 2018a for temperature data by using trend free pre-whitening technique (TFPW), Iqbal et al 2018b for precipitation data which was homogenized and linear increasing trend was found. About Runoff data, Hu et al. 2011 has explained that none of the data series for detecting trend has significant serial correlation at a 5% level.

Interestingly, parametric test (e.g. linear regression method) which is considered as more powerful over non-parmetric (e.g. MK test) but requires data to be distributed normally. By applying linear regression method, trend was found to be positive for precipitation data in the previous study (Iqbal et al. 2018b). Similar result we found in this study also. Meng et al. 2016 has also showed slightly increasing trend in precipitation and decreasing trend of runoff.

Now in revised manuscript we have referred some of these studies about the quality and standard approaches for data analysis in the methodology section. Hope answer will satisfy your concern. The referred studies list is as below;

Von Storch, H., 1999. Misuses of Statistical Analysis in Climate Research. In: Analysis of Climate Variability. Springer, pp. 11–26.

Douglas, E.M.; Vogel, R.M.; Kroll, C.N. Trends in floods and low flows in the United States: Impact of spatial correlation. J. Hydrol. 2000, 240, 90–105, doi:10.1016/S0022-1694(00)00336-X.

Yue, S.; Pilon, P.; Cavadias, G. Power of the Mann-Kendall and Spearman’s rho tests for detecting monotonic trends in hydrological series. J. Hydrol. 2002, 259, 254–271, doi:10.1016/S0022-1694(01)00594-7.

Yuan, F.; Berndtsson, R.; Zhang, L.; Uvo, C.B.; Hao, Z.; Wang, X.; Yasuda, H. Hydro Climatic Trend and Periodicity for the Source Region of the Yellow River. J. Hydrol. Eng. 2015, 20, doi:10.1061/(asce)he.1943-5584.0001182

Iqbal, M.; Wen, J.; Wang, X.; Lan, Y.; Tian, H.; Anjum, M.N.; Adnan, M. Assessment of Air Temperature Trends in the Source Region of Yellow River and Its Sub-Basins, China. Asia-Pacific J. Atmos. Sci. 2018a, 54, 111–123, doi:10.1007/s13143-017-0064-x.

Iqbal, M.; Wen, J.; Wang, S.; Tian, H.; Adnan, M. Variations of precipitation characteristics during the period 1960–2014 in the Source Region of the Yellow River, China. J. Arid Land 2018b, 10, 388–401, doi:10.1007/s40333-018-0008-z.

Hu, Y.; Maskey, S.; Uhlenbrook, S.; Zhao, H. Streamflow trends and climate linkages in the source region of the Yellow River, China. Hydrol. Process. 2011, 25, 3399–3411, doi:10.1002/hyp.8069.

Meng, F.; Su, F.; Yang, D.; Tong, K.; Hao, Z. Impacts of recent climate change on the hydrology in the source region of the Yellow River basin. J. Hydrol. Reg. Stud. 2016, 6, 66–81, doi:10.1016/j.ejrh.2016.03.003.

Q33: Section 3.2 should not be included or should be part of supplementary part because the objective of the study is to quantify impacts of CC and LUC on streamflow, not to find out differences between Land use periods

A33: Thanks for the guidelines. Section 3.2 is somewhat part of study as we studied the impact of change in landuse on runoff at subbasins of the study area and change of landuse is not same in each subbasin (as Figure 6 present this in the revised manuscript). Therefore the change in runoff and ET may can be attributed to different landuse class in each subbasin which will be interesting for the readers and the managers of concerned departments.

Q34: Line 400-413: Just repetition of Table 9, Only write the interesting things or overall results

A34: Thanks for the comment. The said lines were presenting the interpretation of the results given in Table 9. Now we have summarized these lines and keeping in veiw the important things.Hope these chages will meet your requirements.

Q35: Line 415: should be 2010s? better write whole period?

A35: Thanks for the observation. This is figure caption. We have rewritten a part of figure caption in the revised manuscript. Hope this will fulfil your requirement.

Q36: Figure 8: Why this figure is needed here, when figure 7 and Table 9 is given?

A36: Thanks for the observation. Overall these two things providing the same results. Now we have removed the figure from the revised manuscript.

Q37: Table 9: what does R2 and NSE describe, as both look like having same figure?

A37: Thanks for the comment. Statistical and graphical model can be used for the performance evaluation techniques. The quantitative statistics were divided into three major categories: standard regression, dimensionless, and error index as described by Moraise et al. (2007) Standard regression statistics determine the strength of the linear relationship between simulated and measured data. Dimensionless techniques provide a relative model evaluation assessment, and error indices quantify the deviation in the units of the data of interest (Legates and McCabe, 1999).

According to mentioned above, Coefficient of Determination (R2) falls in standard regression category, NSE falls in dimensionless category and PBAIS falls in the category of error index.  

Where, R2 describe the degree of collinearity between simulated an observed and NSE is a normalized statistic that describe the relative magnitude of residual variance or noise compared to the observed data variance. Efforts were made in such a way that we could assess the performance of SWAT model by considering one index from each category. The referred articles are listed below. Hope the answer will meet your requirement.

Moriasi, D.N.; Arnold, J.G.; Van Liew, M.W.; Bingner, R.L.; Harmel, R.D.; Veith, T.L. Model evaluation guidelines for systematic quantification of accuracy in watershed simulations. Trans. ASABE 2007, 50, 885–900.

Legates, D. R., and G. J. McCabe. 1999. Evaluating the use of “goodness-of-fit” measures in hydrologic and hydroclimatic model validation. Water Resources Res. 35(1): 233-241.

Q38: Line 428-429: Not clear, which data Simulated or Observed data.

A38: Thanks for the observations. We have rewritten the text in the revised manuscript and reproduced here also. Hope this will fulfil your requirement.

“In this section, simulated results were adopted instead of observed data to compare the hydrological results in subbasins of the SRYR. Table 9 shows the simulated results of evapotranspiration and runoff by SWAT under four scenarios as presented in section 2.2.3.”

Q39: Figure 9: Better use percentage on y-axis

A39: Thanks for your comment. We have already placed the percentage wise results of Runoff and ET in Table 9. Here in figure 9 we have shown the changes with absolute values to compare the results with other related studies as they also present runoff changes results in absolute values. Hope this will satisfy your concern.

Q40: Line 498-540: What is interesting in these paragraphs? is this according to the objective and thrust of the study? Only discuss the main results and interesting and unique results of the study

A40: Thanks for the guidelines. We have revised the manuscript and removed unnecessary lines and kept most relevant and interesting one. Hope our endeavor will satisfy your requirements.

Q41: Line 545: ET increased due to climate not decrease

A41: Thanks for the observation. We have corrected it in the revised manuscript.

Q42: Line 548-550: there is big difference between this study results and Meng et al. this study 22.4 mm and Meng 6 mm

A42: Thanks for the comment. We have reviewed thoroughly this study and Meng et al. (2016) study results. Meng et al (2016) explained the decreasing runoff trends at the rate of 3.2 mm/10yr, 9.2 mm/10yr and 6.0 mm/10yr, in Jimai, Maqu and Tangnaihai, respectively by analyzing the data for the period of 1961-2013. This show that total runoff in the region is decreasing at the rate of 18.4 mm/decade (summation of all subbasins).

According to our study runoff is also decreasing (Figure 4) at the rate of 36.4 m3/sec per decade (≈ 25 mm/per decade) for the whole catchment by using the data for the period of 1974-2014. So such results can be comparable on the basis of data period.

Now at the said lines (548-550), instead of runoff trends the pattern of magnitude of runoff by climate change in three subbasins are compared which was same as Meng et al (2016). Hope the answer will meet your requirement.

Meng, F.; Su, F.; Yang, D.; Tong, K.; Hao, Z. Impacts of recent climate change on the hydrology in the source region of the Yellow River basin. J. Hydrol. Reg. Stud. 2016, 6, 66–81, doi:10.1016/j.ejrh.2016.03.003.

Reviewer 2 Report

This manuscript needs to following minor modifications:

1- Authors did not state annual minimum, mean and maximum temperature.

2- Table 2, what is DD? North must be mentioned below latitude. Columns 2 and 3 must be displaced for hydrological data.  

3- Authors add new references (from 2018 to 2022) to manuscript such as:

Shokouhifar Y., Lotfirad M., Esmaeili-Gisavandani H., Adib A. (2022) Evaluation of climate change effects on flood frequency in arid and semi-arid basins. Water Supply, In press, https://doi.org/10.2166/ws.2022.271.

Author Response

Comments and Suggestions for Authors

This manuscript needs to following minor modifications:

Q01: Authors did not state annual minimum, mean and maximum temperature.

A01: Thank you for your guideline. We were mentioned about these variable in context of data collection in the revised ms. We hope this will meet your requirement.

Q02: Table 2, what is DD? North must be mentioned below latitude. Columns 2 and 3 must be displaced for hydrological data.  

A02: Thanks for your observations. As per recommendation of another reviewer, the Table 2 was not necessary as all the attributes of stations are defined in Figure 1, so Table 2 was removed from the MS. Hope this will fulfil your requirement.

Q03: Authors add new references (from 2018 to 2022) to manuscript such as:

Shokouhifar Y., Lotfirad M., Esmaeili-Gisavandani H., Adib A. (2022) Evaluation of climate change effects on flood frequency in arid and semi-arid basins. Water Supply, In press, https://doi.org/10.2166/ws.2022.271.

A03: Thanks for your comments. We have revised the MS and added latest references where applicable. Hopefully this will meet your requirement

Reviewer 3 Report

I'd use "for instance" without "an" in line52

Author Response

Comments and Suggestions for Authors

Q1: I'd use "for instance" without "an" in line52

A1: Thanks you for your guideline. We have revised the mentioned line in the revised manuscript, and hope this will meet your requirement.
